# LLM Enhancers for GNNs: An Analysis from the Perspective of Causal Mechanism Identification

**Hang Gao** [* 1 2] **Wenxuan Huang** [* 1 2 3] **Fengge Wu** [1 2 3] **Junsuo Zhao** [1 2 3] **Changwen Zheng** [1 2 3] **Huaping Liu** [4]

## Abstract

The use of large language models (LLMs) as feature enhancers to optimize node representations, which are then used as inputs for graph neural networks (GNNs), has shown significant potential in graph representation learning. However, the fundamental properties of this approach remain underexplored. To address this issue, we propose conducting a more in-depth analysis of this issue based on the interchange intervention method. First, we construct a synthetic graph dataset with controllable causal relationships, enabling precise manipulation of semantic relationships and causal modeling to provide data for analysis. Using this dataset, we conduct interchange interventions to examine the deeper properties of LLM enhancers and GNNs, uncovering their underlying logic and internal mechanisms. Building on the analytical results, we design a plug-and-play optimization module to improve the information transfer between LLM enhancers and GNNs. Experiments across multiple datasets and models validate the proposed module. Codes can be found in `https://github.com/WX4code/LLMEnhCausalMechanism`.

## 1. Introduction

With the rapid development of LLMs (Brown et al., 2020; Devlin et al., 2019; Dubey et al., 2024), their semantic understanding and feature generation capabilities have demonstrated significant potential across various fields (Tian et al., 2024; Mustapha, 2025; Liu et al., 2025). In the domain of graph representation learning, recent studies have integrated LLMs with GNNs to enhance performance (Mao et al.,

2024). One category of methods employs LLMs as feature enhancers to optimize node representations (Chen et al., 2023; Huang et al., 2024a), which are then used as inputs for GNNs to build a unified model. Such methods leverage the pre-trained knowledge of LLMs to generate richer and more semantically coherent features, addressing the limitations of traditional GNNs while incorporating domain knowledge from LLMs into the features (Yu et al., 2023). They also demonstrate strong generalization capabilities in heterogeneous graph representation learning scenarios (Li et al., 2025). Numerous recent methods (Chen et al., 2023; Huang et al., 2023b; Liu et al., 2024) have adopted this *LLM-enhancer-plus-GNN* framework, achieving excellent results across various graph representation learning tasks.

However, despite their broad applications, there is a noticeable lack of dedicated research examining the fundamental framework of the LLM-enhancer-plus-GNN paradigm. The deeper properties and mechanisms underlying this framework remain largely unexplored. This paper aims to address this gap. Meanwhile, achieving this is far from straightforward. Since both the LLM enhancer and GNN are composed of neural networks, each is inherently challenging to model formally (Lu et al., 2024). When these two complex networks are combined, conducting a unified analysis becomes even more difficult.

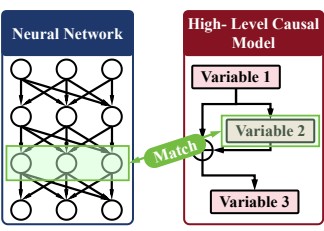

*Figure 1.* A simple illustration of variable alignment achieved by interchange intervention.

To address this issue, we introduce the interchange intervention approach (Wu et al., 2023a; Huang et al., 2023a) from causality theory (Pearl, 2022) to perform the analysis. Specifically, we construct a novel synthetic graph dataset with controllable causal relationships. Here, causal relationships refer to the ground-truth cause-and-effect connections, which can be utilized to evaluate the model's ability to accurately capture and represent key underlying information. While synthetic graph datasets are often used to analyze the properties of graph representation learning algorithms (Wu et al., 2022), our dataset is further enhanced to enable precise manipula-

*Equal contribution [1]Institute of Software, Chinese Academy of Sciences [2]National Key Laboratory of Space Integrated Information System [3]University of Chinese Academy of Sciences [4]Tsinghua University. Correspondence to: Fenge Wu <fengge@iscas.ac.cn>.

tion of complex semantic relationships. This feature makes the dataset more suitable for evaluating the performance and characteristics of the LLM-enhancer-plus-GNN models. Based on this dataset, we generate training and testing data. Using the predefined data synthesis process, we model the intrinsic causal relationships in the data as a high-order causal model, which accurately captures the complex relational structures within the dataset. Subsequently, we train and test the GNN model enhanced by the LLM enhancer using the synthetic data. Through the interchange intervention method, we systematically analyze the correspondence between the LLM-enhancer-plus-GNN model and the high-order causal model, aiming to uncover the internal logical structure of the black-box neural network. Figure 1 provides a simplified illustration of the objectives we aim to achieve. Furthermore, we theoretically validate the reliability of this analytical approach and derive a series of important findings.

Additionally, based on our findings, we identify areas for improvement in the information transfer between the LLM enhancer and GNN. To address this, we developed a plug-and-play optimization module designed to better assist the LLM enhancer in optimizing GNN performance. The effectiveness of this module has been validated across multiple datasets and various models. Our contributions are summarized as follows:

- We construct a synthetic graph dataset with controllable causal relationships, capable of simulating complex semantic associations within graphs.

- We design an analysis framework based on interchange intervention and demonstrate its effectiveness both theoretically and experimentally. Additionally, we also conduct a comprehensive study on the LLM-enhancer-plus-GNN paradigm, yielding a series of findings.

- We propose a novel optimization module to enhance the information transfer between the LLM enhancer and the GNN, validating its effectiveness through experiments on multiple datasets and models.

## 2. Related Works

**Causal Mechanism Identification within Neural Networks.** Research on explaining and understanding deep learning models has been ongoing. For this purpose, some studies attempt to uncover interpretable causal mechanisms within neural networks (Geiger et al., 2020; 2021) and training methods for inducing such interpretable mechanisms (Geiger et al., 2022b; Huang et al., 2023a). These methods can be classified into iterative nullspace projection (Ravfogel et al., 2020; Elazar et al., 2021; Lovering & Pavlick, 2022), causal mediation analysis (Meng et al., 2022; Vig et al., 2020), and causal effect estimation (Abraham et al.,

2022; Elazar et al., 2022; Wu et al., 2023b). A significant body of research has also focused specifically on causal modeling in GNNs or LLMs (Chen et al., 2022; Ko et al., 2023; Gao et al., 2023; 2024; Zhao et al., 2025). We, on the other hand, utilize the causal mechanism identification approach for the specific analysis of the LLM-enhancer-plus-GNN paradigm.

**Enhancing GNNs with LLM.** Using LLM for initial node feature processing followed by GNN to explore inter-node relationships has become a popular research direction (Mao et al., 2024; Fatemi et al., 2023; Chen et al., 2023; Huang et al., 2023b; Liu et al., 2024; Huang et al., 2024a; Tang et al., 2024). Among the methods within this field, some combine prompt learning for graph data enhancement (Liu et al., 2024; He et al., 2024; Tang et al., 2024), others emphasize the use of LLMs for class-level information and application usages (Yu et al., 2023; Ren et al., 2024; Lyu et al., 2023). Our research aims to conduct a thorough evaluation of the overall framework of these methods. Further related works can be found in **Appendix** A.

## 3. Analysis

### 3.1. CCSG Dataset

Our goal is to investigate the LLM-enhancer-plus-GNN paradigm, studying its capabilities in data relationship modeling. To accurately assess such capabilities, it is essential to first identify the relationships present in the dataset. Given the challenges of determining causal relationships in general graph datasets, we have developed a semantically rich graph dataset with controllable internal causal relationships, referred to as the *Controlled Causal-Semantic Graph* (CCSG) dataset.

The CCSG dataset is constructed based on Wikipedia entries, incorporating various types of causal relationships that we have defined. Table 1 presents a comparison between the CCSG dataset and other similar datasets. This includes attributed graph datasets (Citeseer, Wiki-CS), which contain semantic node attributes, as well as synthetic graph datasets (Synthetic Graph, Spurious-Motif, CRCG), which are manually generated datasets that are used for analyzing the properties of GNNs. We provide a detailed introduction below.

#### 3.1.1. NODE ATTRIBUTES

The node attributes of the CCSG dataset include manually generated features and 5,660 Wikipedia entries, organized into three main categories and fifteen subcategories, ensuring diverse information. This dataset forms the foundation for graph construction of CCSG dataset, with detailed descriptions provided in **Appendix** C.

*Table 1.* Comparative analysis of our dataset with other similar datasets. The term "Total Combinations" refers to the maximum possible number of combinations attainable when all available graphical elements are employed and juxtaposed in pairs.

| Dataset | Adjustable Node Attributes | Semantic Node Attributes | Adjustable Edges | Multi-order Relationship Adjustment | Controllable Causal Relationship | Semantic Aware Relationship Manipulation | Total Combinations ($\uparrow$) |
|---|---|---|---|---|---|---|---|
| Citeseer (Kipf & Welling, 2016) | ✗ | ✓ | ✗ | Fixed | ✗ | ✗ | Fixed |
| Wiki-Cs (Wittmann & Fey, 2020) | ✗ | ✓ | ✗ | Fixed | ✗ | ✗ | Fixed |
| Synthetic Graph (Ying et al., 2019) | ✗ | ✗ | ✓ | Adjustable First-order Relationship | ✓ | ✗ | 25 |
| Spurious-Motif (Wu et al., 2022) | ✗ | ✗ | ✓ | Adjustable First-order Relationship | ✓ | ✗ | 36 |
| CRCG (Gao et al., 2024) | ✓ | ✗ | ✓ | Adjustable First-order Relationship | ✓ | ✗ | 3750 |
| CCSG | ✓ | ✓ | ✓ | Adjustable Multi-order Relationship | ✓ | ✓ | 226400 |

### 3.1.2. GRAPH CONSTRUCTION

For graph construction, the CCSG dataset ensures controllable generation of node features, connections, and topological structures. Data generation is controlled in four aspects: **1) Node features**, the semantic information of node features can be actively controlled based on the collected data. **2) Node correlation**, node feature correlations are adjusted based on the collected categories, subclasses, and reference relationships. **3) Topological structure**, diverse topologies are introduced and linked to label data to analyze the impact of graph structure on outcomes. Such controllable graph construction enable the injection of predefined and will formulated causal relationships to be added to the data.

### 3.2. Interchange Intervention Based Evaluation

We analyze the LLM-enhancer-plus-GNN paradigm by evaluating its ability to model predefined causal relationships. To do this, we apply the interchange intervention method (Geiger et al., 2022a) from causal inference. This method treats the neural network as a low-level model and the dataset's causal relationships as a high-level causal model. By modifying the neural network's internal feature representations and comparing them to changes in the high-level causal model, it evaluates the correspondence between the two. This process enables us to explore the underlying mechanisms of the model in greater depth.

### 3.2.1. EVALUATION METHOD OUTLINE

Based on the CCSG dataset, we first construct the data and a high-level causal model, denoted as $h(\cdot)$. The model $h(\cdot)$ represents the underlying causal relationships between the data and the corresponding labels, and it outputs the ground truth label based on the input $G$. Our objective is to employ an interchange intervention to identify which hidden variables in the low-level neural network model $f(\cdot)$—specifically, an LLM enhancer-plus-GNN model—correspond to the variables in $h(\cdot)$.

Specifically, we select two different graph samples: $G^{\text{orig}}$

and $G^{\text{diff}}$. We first input $G^{\text{orig}}$ into the model $h(\cdot)$. Then, we select a variable $Z^h$ within $h(\cdot)$ and replace its value with the value it would take if $G^{\text{diff}}$ were the input. This produces a new output for $h(\cdot)$ where the input remains $G^{\text{orig}}$, but the variable $Z^h$ reflects the state it would have under $G^{\text{diff}}$. We denote the resulting output as $\text{INTINT}(h, G^{\text{orig}}, G^{\text{diff}}, Z^h)$, where $\text{INTINT}(\cdot)$ denotes the conducted interchange intervention operation.

Subsequently, we perform interchange intervention on $f(\cdot)$ using both $G^{\text{orig}}$ and $G^{\text{diff}}$, which is denoted as $\text{INTINT}(f, G^{\text{orig}}, G^{\text{diff}}, Z^f)$. Here, $Z^f$ refers to an internal variable within $f(\cdot)$. Intuitively, if $Z^f$ is the hidden layer variable corresponding to $Z^h$ in the neural network model, then $\text{INTINT}(h, G^{\text{orig}}, G^{\text{diff}}, Z^h)$ and $\text{INTINT}(f, G^{\text{orig}}, G^{\text{diff}}, Z^f)$ should be equal. We compute interchange intervention loss $\mathcal{L}_{\text{II}}$ to measure the discrepancy between them:

$$\mathcal{L}_{\text{II}} = \frac{1}{\mathcal{G}^2} \sum_{G^{\text{orig}} \in \mathcal{G}} \sum_{G^{\text{diff}} \in \mathcal{G}} \mathcal{D}\Big(\text{INTINV}\big(h, G^{\text{orig}}, G^{\text{diff}}, Z^h\big),$$
$$\text{INTINV}\big(f, G^{\text{orig}}, G^{\text{diff}}, Z^f\big)\Big), \qquad (1)$$

where $\mathcal{D}(\cdot)$ represents the metric used to measure the difference between $\text{INTINV}\big(h, G^{\text{orig}}, G^{\text{diff}}, Z^h\big)$ and $\text{INTINV}\big(f, G^{\text{orig}}, G^{\text{diff}}, Z^f\big)$, e.g., for classification tasks, $\mathcal{D}(\cdot)$ can be the cross-entropy loss. The set $\mathcal{G}$ denotes the dataset being utilized. We search for the optimal $Z^f$ that minimizes $\mathcal{L}_{\text{II}}$, which would ensure $Z^f$ in $f(\cdot)$ best aligns with $Z^h$ in $h(\cdot)$. See Section 3.2.3 for justifications. In this way, we can establish the correspondence between the low-level neural network and the high-level causal model.

### 3.2.2. RUNNING EXAMPLE

We present a simplified example to illustrate the application of the interchange intervention-based analysis method. As shown in Figure 2, the process is divided into four steps. Below, we provide a detailed step-by-step explanation of the process.

**Step 1: Data Generation.** In this step, as shown in Figure 2,

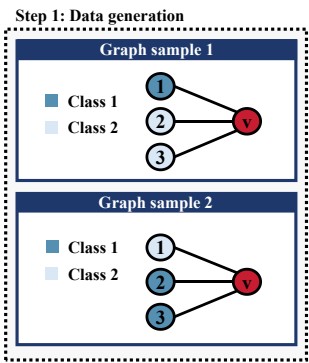
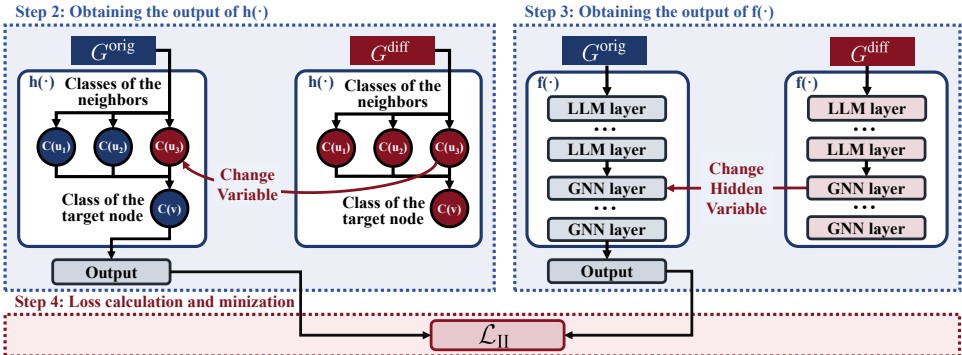

*Figure 2.* The framework of the running example.

we construct only two attributed graph samples, $G_1$ and $G_2$. In practice, the number of samples would be significantly larger. The task is to classify node $v$ depicted in the figure. During data generation, we also define the high-level causal model $h(\cdot)$ for this example. Specifically, we manually establish that the class of node $v$ is determined by first identifying the classes of all its neighboring nodes and then selecting the most frequent class among them as the class of $v$. This logic can be formally expressed as $C(v) = m(\{C(u) \mid u \in \mathcal{N}(v)\})$, where $C(\cdot)$ represents the classes of nodes and $\mathcal{N}(v)$ denotes the set of neighbors of $v$, $m(\cdot)$ selects the most frequent class. Consequently, we have:

$$h(G) = C(v) = m(\{C(u) \mid u \in \mathcal{N}(v)\}). \qquad (2)$$

We then select $C(u_3)$ within $h(\cdot)$ as $Z^h$ for performing the interchange intervention.

**Step 2: Calculating INTINV$(h, G^{\text{orig}}, G^{\text{diff}}, Z^h)$.** We select $G_1$ as $G^{\text{orig}}$ and input it into $h(\cdot)$ to accurately compute the values of various variables and outcomes within the model. We then select $G_2$ as $G^{\text{diff}}$ and input it into $h(\cdot)$, performing another round of calculations for the variables and outcomes. As shown in Figure 2, Step 2, when the input is $G^{\text{orig}}$, we replace the value of $Z^h$, i.e., $C(u_3)$, with its value obtained from $G^{\text{diff}}$ as input and compute the output. This final output represents INTINV$(h, G^{\text{orig}}, G^{\text{diff}}, Z^h)$.

**Step 3: Calculating INTINV$(f, G^{\text{orig}}, G^{\text{diff}}, Z^f)$.** We select $G_1$ as $G^{\text{orig}}$ and input it into $f(\cdot)$. Similarly, we select $G_2$ as $G^{\text{diff}}$ and input it into $f(\cdot)$. Since the optimal selection of $Z^f$ is not yet known, we randomly choose certain hidden layer variables as $Z^f$. We then take the values of $Z^f$ obtained when $G^{\text{diff}}$ is used as input and replace the corresponding values in the model $f(\cdot)$ when $G^{\text{orig}}$ is used as input. The model then outputs INTINV$(f, G^{\text{orig}}, G^{\text{diff}}, Z^f)$.

**Step 4: Loss Calculation and Minimization.** Repeat Step 2 and 3 to acquire all INTINV$(h, G^{\text{orig}}, G^{\text{diff}}, Z^h)$ and INTINV$(f, G^{\text{orig}}, G^{\text{diff}}, Z^f)$ values, then utilize Equation 2 to calculate the loss $\mathcal{L}_{\text{II}}$. We then repeat the process to locate optimal $Z^f$ that minimizes $\mathcal{L}_{\text{II}}$, finding the best alignment of $Z^h$ within $f(\cdot)$, so as to reveal part of the logical structure

within $f(\cdot)$ to facilitate analysis.

### 3.2.3. VALIDATION

To conduct validation of our analysis method, we introduce the concept of *total effect* from causal theory (Pearl, 2009).

**Definition 3.1.** *Total effect* (Pearl, 2009) represents the overall causal impact of one variable $Z$ on another variable $Y$. It is denoted by TE$_{z,z'}(Y)$, where $z$ and $z'$ are two specific values that $Z$ can assume. The total effect TE$_{z,z'}(Y)$ quantifies the expected difference in the outcome $Y$ when $Z$ is set to $z$ compared to when it is set to $z'$.

Therefore, we can formalize our objective more precisely: to find variables $Z^h$ in model $h(\cdot)$ and $Z^f$ in model $f(\cdot)$ such that $Z^h$ and $Z^f$ exhibit consistent total effects on the predicted outputs. As demonstrated in Equation 1, we aim to find the aforementioned variable $Z^f$ by minimizing a loss function $\mathcal{L}_{\text{II}}$. We use the following theorem to prove the validity of this approach.

**Theorem 3.2.** *Given a high-level causal model $h(\cdot)$ and a low-level neural network model $f(\cdot)$, both of which accurately map input graphs $G \in \mathcal{G}$ to outputs $Y \in \mathcal{Y}$, such that $Y = f(G) = h(G)$. Assume there exists a subset $Z^f$ of the intermediate variables in $f(\cdot)$ and a bijective mapping $\eta : Z^f \to Z^h$, where $Z^h$ represents certain variables in $h(\cdot)$. If there exists a $Z^f$ that minimizes the loss $\mathcal{L}_{\text{II}}$, we can conclude that the total effect $TE_{z^f,z^{f'}}(Y^f)$ of $f(\cdot)$ is equal to the total effect $TE_{z^h,z^{h'}}(Y^h)$ of $h(\cdot)$ in all cases. Here, $z^h$ and $z^f$ represent the values of $Z^h$ and $Z^f$ respectively, given the same input graph $G$. Similarly, $z^{h'}$ and $z^{f'}$ represent their values for a different input graph $G'$.*

The demonstration can be found in **Appendix** B.1. Theorem 3.2 establishes that, under the condition where the model $f(\cdot)$ performs sufficiently well and a bijection exists between the internal variables of $f(\cdot)$ and the internal variable $Z^h$ of $h(\cdot)$, minimizing $\mathcal{L}_{\text{II}}$ yields a $Z^f$ whose total effects on the model outputs are exactly the same as those of the corresponding $Z^h$. Equal total effects indicate that the neurons modeling the variable $Z^h$ have been precisely identified,

thereby providing theoretical validation for the effectiveness of the proposed method. However, if the aforementioned bijection does not exist, how will the results change? The following corollary provides further analysis.

**Corollary 3.3.** *Even if the condition that there exists a subset $Z^f$ of the intermediate variables of $f(\cdot)$ satisfied $\eta : Z^f \to Z^h$ where $\eta$ is bijective does not hold, if $INTINV\big(f, G^{orig}, G^{diff}, Z^f\big) = INTINV\big(h, G^{orig}, G^{diff}, Z^h\big)$ holds, the conclusion given in Theorem 3.2 remains valid.*

The demonstration can be found in **Appendix B.2**. Corollary 3.3 shows that even if the aforementioned bijection does not exist, the condition $INTINV\big(f, G^{orig}, G^{diff}, Z^f\big) = INTINV\big(h, G^{orig}, G^{diff}, Z^h\big)$ ensures the existence of a $Z^f$ that fully corresponds to $Z^h$. In practical analysis, we can use the distance of $\mathcal{L}_{II}$ from its possible minimum value to assess the degree of matching between $Z^f$ and $Z^h$.

**Proposition 3.4.** *For a high-level causal model $h(\cdot)$ and a low-level neural network $f(\cdot)$, both mapping input graphs $G \in \mathcal{G}$ to outputs $Y \in \mathcal{Y}$ such that $Y = f(G) = h(G)$, if $Z^f$ within $f(\cdot)$ and $Z^h$ within $h(\cdot)$ minimize $\mathcal{L}_{II}$ to its optimal value $\mathcal{L}_{II}^*$, then $Z^f$ aligns best with $Z^h$ and has an identical total effect on the output prediction as $Z_h$. The minimal $\mathcal{L}_{II}^*$ is given by:*

$$\mathcal{L}_{II}^* = \frac{1}{\mathcal{G}^2} \sum_{G^{orig} \in \mathcal{G}} \sum_{G^{diff} \in \mathcal{G}} \mathcal{D}\Big( INTINV\big(h, G^{orig}, G^{diff}, Z^h\big), \\ INTINV\big(h, G^{orig}, G^{diff}, Z^h\big)\Big). \quad (3)$$

Please refer to **Appendix B.3** for the justification. In our experiments, we use cross-entropy loss as $\mathcal{D}(\cdot)$, and the label probabilities predicted by $h(\cdot)$ are either 0 or 1. Therefore, in our experiments, $\mathcal{L}_{II}^* = 0$. Consequently, we directly use $\mathcal{L}_{II}$ to represent the distance between $\mathcal{L}_{II}$ and $\mathcal{L}_{II}^*$.

### 3.3. Analytical Experiments

We analyze the LM-enhancer-plus-GNN framework using the CCSG dataset and the approach from Section 3.2, looking for deeper insights and possible optimization of the framework. Following prior studies on causality in graph representation learning (Wu et al., 2022; Gao et al., 2024), we treat nodes as the smallest variable units, focusing on node relationships without modeling internal node interactions. Most approaches use fixed-parameter LLM enhancers to reduce training costs (Liu et al., 2024; Huang et al., 2023b), our experiments also follow this setting.

**Theorem 3.5.** *Given a high-level causal model $h(\cdot)$ and a variable $\bar{Z}^h$ within $h(\cdot)$, suppose there exists a variable $\bar{Z}^f$ within $f(\cdot)$, where $f(\cdot)$ is a GNN with LLM enhancers,*

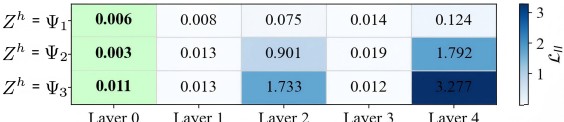

(a) The results of node-level experiments where $Z^h$ is set to the output variables $\Psi_1$, $\Psi_2$, and $\Psi_3$ of $h^{node,1}(\cdot)$.

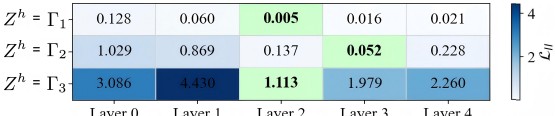

(b) The results of node-level experiments where $Z^h$ is set to the output variables $\Phi_1$ and $\Phi_2$ of $h^{node,2}(\cdot)$.

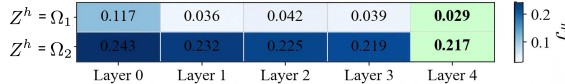

(c) The results of graph-level experiments where $Z^h$ is set to the output variables $\Gamma_1$, $\Gamma_2$, and $\Gamma_3$ of $h^{graph,1}(\cdot)$.

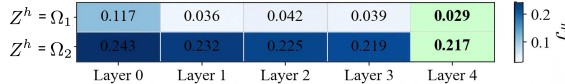

(d) The results of graph-level experiments where $Z^h$ is set to the output variables $\Omega_1$ and $\Omega_2$ of $h^{graph,2}(\cdot)$.

*Figure 3.* Analytical experiment results. In the figure, each small square represents the optimal value of $\mathcal{L}_{II}$ for a specific GNN layer. Green squares indicate the minimum value of $\mathcal{L}_{II}$ within the corresponding row. Details concerning the variables can be found in **Appendix C**.

*satisfying the following condition:*

$$INTINV\big(f, G^{orig}, G^{diff}, \bar{Z}^f\big) = INTINV\big(h, G^{orig}, G^{diff}, \bar{Z}^h\big), \quad (4)$$

*where $G^{orig}$ and $G^{diff}$ differ at or above the scale of individual nodes. Then, the internal variables within the GNN model are always sufficient to constitute $\bar{Z}^f$.*

The proof can be found in **Appendix B.4**. According to the theorem, and taking into account the difficulty of fixed LLMs in modeling causal relationships, the subsequent analysis focuses on using the GNN model to examine causal relationship modeling within the framework.

#### 3.3.1. NODE-LEVEL ANALYSIS

We begin our analysis with node-level tasks by first constructing a node classification dataset that leverages the rich information in the CCSG and designing $h^{node}(\cdot)$. $h^{node}(\cdot)$ outputs the class of a target node $v$. Each graph sample is used to classify only one target node $v$. This ensures the elimination of potential mutual influences that may arise when multiple nodes are classified simultaneously, which could prevent $h^{node}(\cdot)$ from being accurately established.

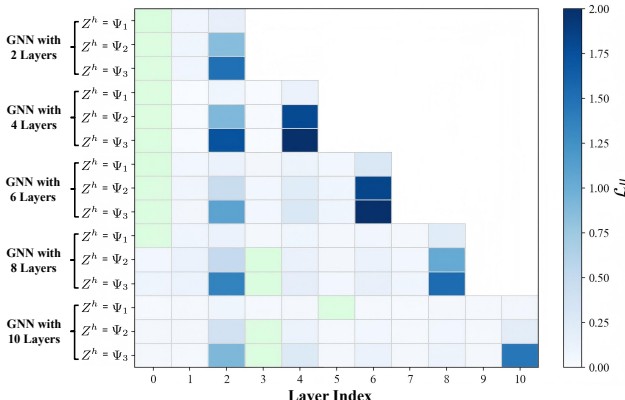

(a) The results from experiments where $Z^h$ is set to the output variables of $h^{\text{node},1}(\cdot)$, with different number of GNN layers.

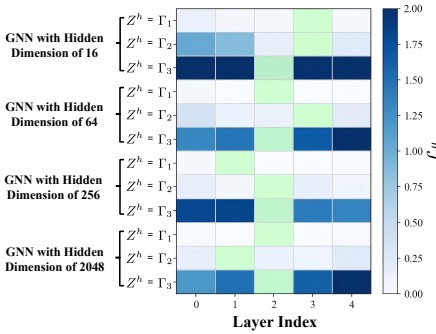

(b) The results from experiments where $Z^h$ is set to the output variables of $h^{\text{graph},1}(\cdot)$, with different hidden dimension.

*Figure 4.* Experimental results under different GNN scales.

The formal definition of $h^{\text{node}}$ is as follows:

$$h^{\text{node}}(G) = h^{\text{node},3} \circ h^{\text{node},2} \circ h^{\text{node},1}(G), \qquad (5)$$

Here, the symbol $\circ$ denotes function composition. The function $h^{\text{node},1}(\cdot)$ is responsible for processing single node feature, while $h^{\text{node},2}(\cdot)$ captures and analyzes the relationships and properties among these nodes. Finally, $h^{\text{node},3}(\cdot)$ serves as the ultimate processing function, outputting the features of the target node $v$. The details can be found in **Appendix** C. To ensure generalizability, we construct our LLM-enhancer-plus-GNN model based on commonly used GNNs and LLMs. Specifically, the GNN module is implemented using GCN (Kipf & Welling, 2016) and LLM enhancer is built using Llama 3 (Dubey et al., 2024). To ensure precise analysis, we adjust the number of training epochs and modify the task difficulty so that all models achieve an accuracy of at least 90%.

The results are presented in Figures 3(a) and 3(b). It can be observed that the best alignment of variables (marked in green) across different node levels occurs in layer 0 and layer 1. These positions can be regarded as the locations of the internal representations $Z^h$ within the GNN model. Additionally, it can also be observed that the optimal align-

ment for $h^{\text{node},1}(\cdot)$ is positioned slightly ahead of the optimal alignment for $h^{\text{node},2}(\cdot)$.

### 3.3.2. GRAPH-LEVEL ANALYSIS

For graph-level task analysis, we manually set the utilized topological structures, node features, and their interrelations, which collectively form the graph sample and determine the label. The corresponding high-level causal model $h^{\text{graph}}$ can be formulated as follows:

$$h^{\text{graph}}(G) = h^{\text{graph},3} \circ h^{\text{graph},2} \circ h^{\text{graph},1}(G), \qquad (6)$$

where $h^{\text{graph},1}(\cdot)$ processes the features of substructures, $h^{\text{graph},2}(\cdot)$ denotes the function locating the topological structures, and $h^{\text{graph},3}(\cdot)$ denotes the final processing function that outputs the graph label. The details can be found in **Appendix** C.

The results are presented in Figure 3(c) and Figure 3(d). Similar to the phenomenon observed in the node-level experiments, the optimal alignment for $h^{\text{graph},1}(\cdot)$ is positioned ahead of the optimal alignment for $h^{\text{graph},2}(\cdot)$.

Based on the experimental results at both the node-level and graph-level, it can be found that when $Z^h$ is more strongly associated with node-level information and is closer to the raw input data in the logical structure of the model $h(\cdot)$, its corresponding latent variable $Z^f$ is more likely to appear in the shallow layers of the GNN structure. This indicates that the role of the LLM enhancer is to process node-level and raw data-level information. Such a finding also partially validates Theorem 3.5. In summary, this conclusion can be further generalized into the following empirical finding.

**Empirical Finding 1.** *For fixed-parameter LLM enhancers, the features output by the LLM serve the function of representing information at the node level and the raw data level.*

### 3.3.3. DEEPER INSIGHT

For further analysis, we repeated the experiments under different model scales by increasing the number of layers and the hidden dimensions of the GNN. The results are shown in Figure 4. The experiments reveal an intriguing phenomenon: *for both node-level and graph-level variables, the alignment results of $Z^h$ within the GNN exhibit a certain regularity across different model scales.* For example, as shown in Figure 4, despite variations in model depth, the alignment performance is consistently worse for even-numbered layers than for odd-numbered layers, with the worst performance observed in the final layer. Furthermore, for $Z^h = \Phi_3$, the performance remains suboptimal across all settings. Figure 4(b) illustrates that regardless of the size of the hidden dimensions, the optimal alignment results for $Z^h$ consistently

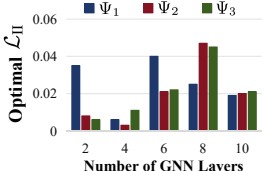
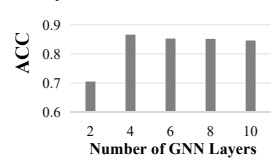
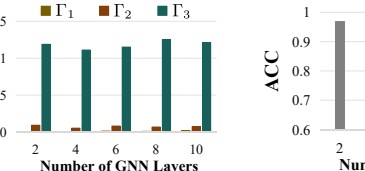
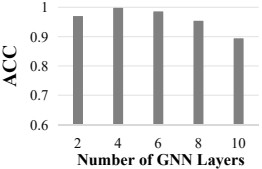

(a) Node-level task alignment results with different number of GNN Layers.

(b) Node-level task model accuracy with different number of GNN Layers.

(c) Graph-level task alignment results with different number of GNN Layers.

(d) Graph-level task model accuracy with different number of GNN Layers.

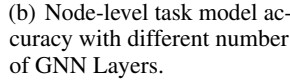
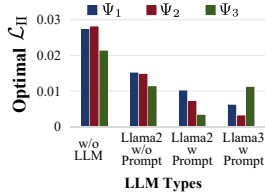
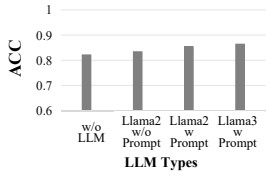
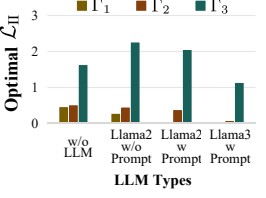
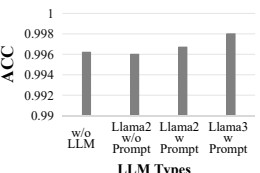

(e) Node-level task alignment results with different LLMs.

(f) Node-level task model accuracy with different LLMs.

(g) Graph-level task alignment results with different LLMs.

(h) Graph-level task model accuracy with different LLMs.

*Figure 5.* Experimental results under different model scales and the use of LLMs.

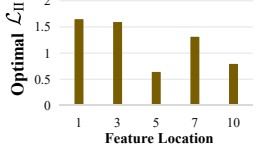
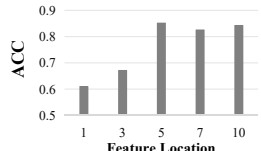

(a) Node-level task alignment results with different feature positions.

(b) Node-level task model accuracy with different feature positions.

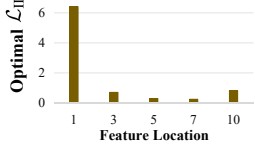
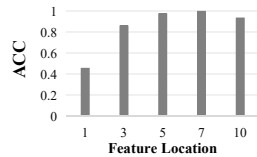

(c) Graph-level task alignment results with different feature positions.

(d) Graph-level task model accuracy with different feature positions.

*Figure 6.* Experimental results under different feature positions. The numbers represent the relative positions of the tokens corresponding to the adopted features within the entire output, ranging from a minimum of 1 to a maximum of 10.

occur in the middle layers. In **Appendix** E, we present numerous additional experiments, all of which corroborate the existence of this phenomenon. This phenomenon leads to the following conclusion:

**Empirical Finding 2.** *After receiving input from the LLM enhancer, the neural structure within the GNN exhibits a relatively consistent logical pattern, maintaining a certain degree of invariance despite changes in the model's scale.*

For further exploration, experiments were performed to compare alignment with accuracy. To facilitate analysis,

accuracy is no longer forced to be above 90%. Within the results in Figure 5, it can be observed that there is a certain correlation between the optimal alignment, as represented by $\mathcal{L}_{II}$, and the model's performance.

**Empirical Finding 3.** *The analysis based on $\mathcal{L}_{II}$ can partially reflect the capability of the model. Specifically, a lower optimal $\mathcal{L}_{II}$ value generally indicates stronger model capability and vice versa.*

Extra experiments within **Appendix** E indicate the same. Combining with Empirical Finding 2, we hypothesize that the scaling up of the GNN only enlarges its structure, without enhancing its capacity for causal relation modeling. Meanwhile, Figures 5(e), 5(f), 5(g), and 5(h) demonstrate that more powerful LLMs enhance model performance. Yet, improving the LLM backbone remains challenging due to the high resource cost. Empirical Finding 1 shows that the LLM enhancer in the architecture primarily provides information at both the node level and the raw data level. Thus, we instead shift our focus to the optimization of the connection between the LLM enhancer and GNN.

We modified the features transmitted from the LLM enhancer to the GNN and conducted experiments to assess their impact. Specifically, while conventional methods (Chen et al., 2023; Huang et al., 2023b; Liu et al., 2024) typically select last-layer features corresponding to specific token positions in the LLM enhancer output as input for the subsequent GNN, we varied these positions to alter the transmitted information. Experimental results, as shown in Figures 6, demonstrate that changes in the selection of these positions significantly affect model performance, with a greater impact than the previously discussed factors. As a

*Figure 7.* Framework of the proposed AT module.

*Table 2.* Performance comparison across different backbone 'matches of LLMs and GNNs. **Diff** denotes performance change achieved with AT module.

| Dataset | Method | GCN | | | GAT | | | GraphSAGE | | |
|---|---|---|---|---|---|---|---|---|---|---|
| | | w/o AT | w/ AT | **Diff** | w/o AT | w/ AT | **Diff** | w/o AT | w/ AT | **Diff** |
| Cora | Llama2 | 83.71 ± 1.05 | 84.67 ± 0.79 | +0.96 | 83.67 ± 1.34 | 84.47 ± 0.43 | +0.80 | 84.45 ± 0.98 | 85.12 ± 1.14 | +0.67 |
| | Qwen2 | 84.33 ± 0.74 | 86.36 ± 0.35 | +2.03 | 84.55 ± 0.77 | 86.13 ± 0.87 | +1.58 | 84.11 ± 0.57 | 85.34 ± 0.75 | +1.23 |
| | Llama3 | 84.98 ± 1.00 | 86.74 ± 0.32 | +1.76 | 84.87 ± 0.85 | 86.29 ± 0.21 | +1.42 | 85.13 ± 0.76 | 86.43 ± 0.22 | +1.30 |
| Pubmed | Llama2 | 81.18 ± 0.46 | 83.82 ± 0.60 | +2.64 | 81.86 ± 0.84 | 83.86 ± 0.42 | +2.00 | 81.82 ± 1.51 | 84.77 ± 0.57 | +2.95 |
| | Qwen2 | 81.09 ± 0.34 | 84.04 ± 1.61 | +2.95 | 81.45 ± 0.29 | 83.11 ± 0.14 | +1.66 | 81.88 ± 1.29 | 84.35 ± 1.64 | +2.47 |
| | Llama3 | 82.06 ± 1.35 | 84.43 ± 1.42 | +2.37 | 82.48 ± 0.88 | 84.86 ± 1.60 | +2.38 | 82.23 ± 0.79 | 85.32 ± 1.49 | +3.09 |
| Instagram | Llama2 | 62.81 ± 0.98 | 64.13 ± 0.82 | +1.32 | 63.12 ± 0.75 | 64.82 ± 0.42 | +1.70 | 64.55 ± 1.73 | 66.78 ± 0.43 | +2.23 |
| | Qwen2 | 63.12 ± 1.45 | 64.45 ± 0.98 | +1.33 | 62.81 ± 1.44 | 64.76 ± 1.18 | +1.95 | 64.48 ± 0.87 | 66.23 ± 0.77 | +1.75 |
| | Llama3 | 63.55 ± 0.43 | 64.86 ± 0.22 | +1.31 | 63.64 ± 0.23 | 65.51 ± 0.10 | +1.87 | 65.40 ± 0.15 | 67.17 ± 0.24 | +1.77 |

result, we chose to optimize the model based on the selection of these positions.

# 4. Method

Based on the previous experimental analysis, we propose optimizing the data transmission between the LLM enhancer and the GNN by introducing the *Attention-based Transmission* (AT) module. The design of the AT module is simple and straightforward, primarily leveraging an attention mechanism to select the most optimal parts of the information output by the LLM enhancer for downstream transmission, thereby optimizing the model. Figure 7 illustrates the framework of the module.

Specifically, to conduct a more comprehensive feature search, we first use the LLM to generate $q$ different input prompts for the LLM enhancer, where $q$ is a hyperparameter. As a result, we obtain $q$ different feature sets, $X^1, X^2, \ldots, X^q$. Each $X^i$ contains the features corresponding to the output tokens from the LLM enhancer using the $i$-th prompt. Specifically, $X^i = \{x^i_j\}_{j=1}^n$, where $x^i_j$ denotes the feature of the $j$-th token from the final layer of the LLM enhancer when the $i$-th prompt is provided as input.

Since the number $n$ of output tokens is not fixed and varies with different inputs, we select a total of $m$ feature vectors, where $m$ is a hyperparameter. The indices of the selected vectors are determined based on $n$ and $m$, and are computed as follows: $\{\lfloor \min(1 \times \frac{n}{m}, 1) \rfloor, \lfloor \min(2 \times \frac{n}{m}, 1) \rfloor, \ldots, \lfloor \min(m \times \frac{n}{m}, 1) \rfloor\}$. We then select the corresponding features and denote them as $S^i = \{s^i_j\}_{j=1}^m$.

Next, we input $S^1, S^2, \ldots, S^q$ into a transformer encoder

separately to generate attention scores. Specifically, for $S^i$, we acquire attention matrix $A^i$:

$$A^i = Q^i(K^i)^\top, \qquad (7)$$

where $Q^i$ and $K^i$ denote the output Query and Key matrices of the final layer of the transformer encoder. Then, compute the average attention scores for vector $\boldsymbol{\alpha}^i \in \mathbb{R}^m$, the $j$-th element of $\boldsymbol{\alpha}^i$ can be represented as:

$$\boldsymbol{\alpha}^i_j = \frac{1}{m} \sum_{l=1}^m A^i_{jl}, \quad \text{for } j = 1, 2, \ldots, m. \qquad (8)$$

we acquire $q$ vectors, $\boldsymbol{\alpha}^1, \boldsymbol{\alpha}^2, \ldots, \boldsymbol{\alpha}^q$. Next, we applied the softmax function to normalize all the elements within all vectors together and acquired normalized results $\bar{\boldsymbol{\alpha}}^1, \bar{\boldsymbol{\alpha}}^2, \ldots, \bar{\boldsymbol{\alpha}}^q$. Then, calculate the final output vector $\boldsymbol{z}$ of the LLM enhancer:

$$\boldsymbol{z} = \frac{1}{qm} \sum_{i=1}^q \sum_{j=1}^m \bar{\boldsymbol{a}}^i_j \boldsymbol{s}^i_j, \qquad (9)$$

where $\boldsymbol{z}$ will be utilized as the node feature and input into the GNN. We utilize $\delta$ epochs for training the selection of the prompts, after $\delta$ epochs, we fix the utilized prompt and remove the rest.

We validated our module on Cora (Kipf & Welling, 2016), Pubmed (Hamilton et al., 2017), and Instagram (Huang et al., 2024b) datasets. We utilize Llama2 (Touvron et al., 2023), Qwen2 (Yang et al., 2024a), and Llama3 (Dubey et al., 2024) as the LLM backbones, and GCN (Kipf & Welling, 2016), GAT (Velickovic et al., 2018), and GraphSAGE (Hamilton et al., 2017) as the GNN backbones. The results are shown in Table 2. It can be observed that the AT module is effective across various LLM frameworks. All experimental settings and details are provided in **Appendix D**.

# 5. Conclusion

This paper analyzes the LLM-enhancer-plus-GNN paradigm using the CCSG dataset and interchange intervention. The proposed method is validated through theory and experiments. Additionally, a novel AT module is proposed to optimize the paradigm further.

# Impact Statement

This paper presents work whose goal is to advance the field of Graph Representation Learning and LLM. There are many potential societal consequences of our work, none which we feel must be specifically highlighted here.

# Acknowledgments

We would like to express our sincere gratitude to the reviewers of this paper, as well as the Program Committee and Area Chairs, for their valuable comments and suggestions. This work is supported by the CAS Project for Young Scientists in Basic Research, Grant No. YSBR-040 and Application Innovation Plan, Project No. 825DBB3C

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

# A. Extended Related Works

## A.1. Causal Abstraction within Neural Networks

Causal abstraction, originating from research in causal theory, involves refining and summarizing the causal relationships present in a low-level model into a high-level causal model (Iwasaki & Simon, 1994; Chalupka et al., 2017; Rubenstein et al., 2017). This process ensures that both models exhibit the same causal effects under soft or hard interventions (Massidda et al., 2023). Achieving a fully precise causal abstraction is challenging; therefore, the more commonly used and observed approach is the approximate causal abstraction. Recent research on causal abstraction have been widely applied within neural networks, which can be broadly classified into three key approaches: iterative nullspace projection, causal mediation analysis, and causal effect estimation (Beckers et al., 2019).

Iterative nullspace projection (Ravfogel et al., 2020; Elazar et al., 2021; Lovering & Pavlick, 2022) leverages linear transformations on a model's representation space to project certain attributes into the nullspace, effectively removing their influence. This technique has been employed to identify the role of specific features in model decisions. For example, in (Ravfogel et al., 2020), the authors applied nullspace projection to remove gender-related information from word embeddings to assess its impact on downstream tasks. Other works (Elazar et al., 2021) extended this to different fairness-related contexts, assessing bias removal and its impact on neural network performance.

Causal mediation analysis (Vig et al., 2020; Meng et al., 2022) examines how an intermediary variable transmits the effect from one variable to another. Notably, (Vig et al., 2020) applied this framework to interpret attention mechanisms in Transformer models, showing how specific tokens influence model predictions through attention heads.

Causal effect estimation (Abraham et al., 2022; Elazar et al., 2022; Wu et al., 2023b) focuses on quantifying the causal impact of changing specific input variables on the output. For example, (Abraham et al., 2022) explored how modifying specific neurons in a neural network can alter model predictions, allowing the identification of neurons with strong causal influence over particular tasks. Due to the complexity of large language models (LLMs), the relationships they model are easily influenced by various factors (Yang et al., 2023; 2024b), making their mechanisms more difficult to understand. Therefore, exploring their internal causal mechanisms represents a feasible approach to clarifying their underlying workings. (Wu et al., 2023b) demonstrated how interventions could uncover causal dependencies in integrated architectures like GNNs combined with LLMs, offering insights into their interaction and decision-making mechanisms. On the other hand, Interchange Intervention Training (IIT) (Geiger et al., 2022b) based methods achieve causal abstraction by aligning the causal effects between high-level and low-level models through the use of interchange interventions (Wu et al., 2023a; Huang et al., 2023a). These methods have demonstrated excellent results when applied to neural networks (Geiger et al., 2024).

These methods offer valuable tools for understanding the internal causal structure of complex deep learning models, providing interpretable insights into their operation. We build on this foundation to explore the causal mechanisms in GNNs integrated with LLMs by applying these methodologies to elucidate the interactions between their components and uncover their causal dependencies.

## A.2. LLM as GNN Enhancers

With the emergence of LLMs (Brown et al., 2020; Devlin et al., 2019; Dubey et al., 2024), a new research direction has gained popularity in the field of graph representation learning: using LLMs for initial node feature processing, followed by GNNs for in-depth exploration and utilization of inter-node relationships. This combined approach is expected to address more complex graph data analysis problems effectively (Mao et al., 2024).

In this domain, related methods focus on optimizing the prompts used with LLMs and integrating the more flexible data processing capabilities that LLMs offer. This integration results in graph data that is more readily analyzable by subsequent GNN processes, allowing for better summarization and analysis of data features (Huang et al., 2024a; Chen et al., 2023). Among these methods, most approaches focus on the utilization of LLMs to enhance the interpretation and alignment of graph data. For instance, TAPE (He et al., 2024) prompts an LLM to perform zero-shot classification, requests textual explanations for its decision-making process, and designs an LLM-to-LM interpreter to translate these explanations into informative features for downstream GNNs. Similarly, OFA (Liu et al., 2024) represents both graph data and tasks as nodes, aligning different textual contents into feature vectors of the same dimension. This alignment enables a consistent representation of data, which is then processed using GNN. By ensuring that all data, regardless of its initial format, can be uniformly processed, OFA improves the overall effectiveness of graph data analysis. CasMLN (Wang et al., 2024) also

follows this trend by introducing LLM-based external knowledge to effectively capture the implicit nature of graphs and node types, thereby enhancing type- and graph-level representations.

Some others utilize LLM to enhance class-level information. e.g., ENG (Yu et al., 2023) leverages LLMs to enhance class-level information and seamlessly introduces labeled nodes and edges without altering the raw dataset, facilitating node classification tasks in few-shot scenarios. Additionally, some works (Ren et al., 2024; Lyu et al., 2023) optimize node attribute information using LLMs in specific recommendation tasks, followed by subsequent graph learning.

All these methods follow the general framework of using LLMs for initial data processing, making the data more amenable to analysis, and then utilizing GNNs for a deeper investigation into the relationships and structures within the data. Our research focuses on analyzing and expanding upon this overall framework, aiming to improve the integration and effectiveness of LLM and GNN combinations in graph representation learning.

# B. Proofs

## B.1. Proof of Theorem 3.2.

**Theorem 3.2.** *Given a high-level causal model $h(\cdot)$ and a low-level neural network model $f(\cdot)$, both of which accurately map input graphs $G \in \mathcal{G}$ to outputs $Y \in \mathcal{Y}$, such that $Y = f(G) = h(G)$. Assume there exists a subset $Z^f$ of the intermediate variables in $f(\cdot)$ and a bijective mapping $\eta : Z^f \to Z^h$, where $Z^h$ represents certain variables in $h(\cdot)$. If there exists a $Z^f$ that minimizes the loss $\mathcal{L}_{II}$, we can conclude that the total effect $TE_{\mathbf{z}^f, \mathbf{z}^{f'}}(Y^f)$ of $f(\cdot)$ is equal to the total effect $TE_{\mathbf{z}^h, \mathbf{z}^{h'}}(Y^h)$ of $h(\cdot)$ in all cases. Here, $\mathbf{z}^h$ and $\mathbf{z}^f$ represent the values of $Z^h$ and $Z^f$, respectively, for the same input graph $G$. Similarly, $\mathbf{z}^{h'}$ and $\mathbf{z}^{f'}$ represent their values for a different input graph $G'$.*

To support the proof of Theorem 3.2, we first present the following lemma and provide the corresponding proof.

**Lemma B.1.** *Given the conditions within Theorem 3.2, if $\mathcal{L}_{II}$ reaches minimal, then for all $G^{orig} \in \mathcal{G}$ and $G^{diff} \in \mathcal{G}$, the following equation holds:*

$$INTINV(f, G^{orig}, G^{diff}, Z^f) = INTINV(h, G^{orig}, G^{diff}, Z^h). \tag{10}$$

*Proof.* According to Equation 1, $\mathcal{L}_{II}$ can be represented as:

$$\mathcal{L}_{II} = \sum_{G^{orig} \in \mathcal{G}} \sum_{G^{diff} \in \mathcal{G}} \mathcal{D}\Big(INTINV\big(h, G^{orig}, G^{diff}, Z^h\big), INTINV\big(f, G^{orig}, G^{diff}, Z^f\big)\Big), \tag{11}$$

as $\mathcal{D}(\cdot)$ denotes difference between $INTINV(f, G^{orig}, G^{diff}, Z^f)$ and $INTINV(h, G^{orig}, G^{diff}, Z^h)$, we have:

$$\begin{aligned}
\mathcal{L}_{II} &= \sum_{G^{orig} \in \mathcal{G}} \sum_{G^{diff} \in \mathcal{G}} \mathcal{D}\Big(INTINV\big(h, G^{orig}, G^{diff}, Z^h\big), INTINV\big(f, G^{orig}, G^{diff}, Z^f\big)\Big) \\
&\geq \sum_{G^{orig} \in \mathcal{G}} \sum_{G^{diff} \in \mathcal{G}} \mathcal{D}\Big(INTINV\big(h, G^{orig}, G^{diff}, Z^h\big), INTINV\big(h, G^{orig}, G^{diff}, Z^h\big)\Big),
\end{aligned} \tag{12}$$

$\sum_{G^{orig} \in \mathcal{G}} \sum_{G^{diff} \in \mathcal{G}} \mathcal{D}\Big(INTINV\big(h, G^{orig}, G^{diff}, Z^h\big), INTINV\big(h, G^{orig}, G^{diff}, Z^h\big)\Big)$ is the minimal possible value for $\mathcal{L}_{II}$, where Equation 10 holds. Next, we demonstrate step by step that there exists a certain set of $Z^f$ that allows $\mathcal{L}_{II} = \sum_{G^{orig} \in \mathcal{G}} \sum_{G^{diff} \in \mathcal{G}} \mathcal{D}\Big(INTINV\big(h, G^{orig}, G^{diff}, Z^h\big), INTINV\big(h, G^{orig}, G^{diff}, Z^h\big)\Big)$. Given the conditions within Theorem 3.2, there exists a subset $Z^f$ of the intermediate variables of $f(\cdot)$ that satisfied $\eta(Z^f) = Z^h$. As $\eta$ is bijective, we can find $Z^f$ that satisfied $\eta^{-1}(Z^h) = Z^f$, furthermore, $\eta(Z^f) = \eta(\eta^{-1}(Z^h)) = Z^h$. Then, by setting $Z^f = \eta^{-1}(Z^h)$, we have:

$$INTINV(f, G^{orig}, G^{diff}, Z^f) = f^{latter}\Big(\big(f^{pre}(G^{orig}) \setminus f^{Z^f}(G^{orig})\big) \cup \eta^{-1}\big(h^{Z^h}(G^{diff})\big)\Big), \tag{13}$$

Where $f^{pre}(\cdot)$ outputs all the intermediate variables of the hidden layers corresponding to $Z^f$, and $f^{Z^f}(G^{orig})$ denotes the value of $Z^f$ given the input $G^{orig}$. The function $f^{latter}(\cdot)$ represents the output of the model $f(\cdot)$ given the intermediate

variables output by $f^{\text{pre}}(\cdot)$, and $h^{Z^h}(G^{\text{diff}})$ denotes the value of $Z^h$ given $G^{\text{diff}}$ as input. Furthermore, given the conditions within Theorem 3.2, we have:

$$f^{\text{latter}}\left(f^{\text{pre}}(G^{\text{orig}})\right) = h\left(G^{\text{orig}}\right), \tag{14}$$

therefore:

$$f^{\text{latter}}\left(f^{\text{pre}}(G^{\text{orig}}) \setminus f^{Z^f}(G^{\text{orig}}) \cup f^{Z^f}(G^{\text{orig}})\right) = h^{\text{latter}}\left(h^{\text{pre}}(G^{\text{orig}}) \setminus h^{Z^h}(G^{\text{orig}}) \cup h^{Z^h}(G^{\text{orig}})\right), \tag{15}$$

Where $h^{\text{pre}}(\cdot)$ outputs the intermediate variables within $h(\cdot)$ up to and including $Z^h$, and $h^{\text{latter}}(\cdot)$ denotes the subsequent computation of $h(\cdot)$ that produces the final output given the output of $h^{\text{pre}}(\cdot)$. Furthermore, we have:

$$f^{\text{latter}}\left(f^{\text{pre}}(G^{\text{orig}}) \setminus f^{Z^f}(G^{\text{orig}}) \cup \eta^{-1}\left(h^{Z^h}(G^{\text{orig}})\right)\right) = h^{\text{latter}}\left(h^{\text{pre}}(G^{\text{orig}}) \setminus h^{Z^h}(G^{\text{orig}}) \cup h^{Z^h}(G^{\text{orig}})\right). \tag{16}$$

Then, we have:

$$\begin{aligned}
\text{INTINV}\left(f, G^{\text{orig}}, G^{\text{diff}}, Z^f\right) &= f^{\text{latter}}\left(f^{\text{pre}}(G^{\text{orig}}) \setminus f^{Z^f}(G^{\text{orig}}) \cup \eta^{-1}\left(h^{Z^h}(G^{\text{diff}})\right)\right) \\
&= h^{\text{latter}}\left(h^{\text{pre}}(G^{\text{orig}}) \setminus h^{Z^h}(G^{\text{orig}}) \cup h^{Z^h}(G^{\text{diff}})\right) \\
&= \text{INTINV}\left(h, G^{\text{orig}}, G^{\text{diff}}, Z^h\right).
\end{aligned} \tag{17}$$

Therefore, we can conclude that the minimal possible value of $\mathcal{L}_{\text{II}}$ which is $\sum_{G^{\text{orig}} \in \mathcal{G}} \sum_{G^{\text{diff}} \in \mathcal{G}} \mathcal{D}\left(\text{INTINV}\left(h, G^{\text{orig}}, G^{\text{diff}}, Z^h\right), \text{INTINV}\left(h, G^{\text{orig}}, G^{\text{diff}}, Z^h\right)\right)$ can be reached, where $\text{INTINV}\left(f, G^{\text{orig}}, G^{\text{diff}}, Z^f\right) = \text{INTINV}\left(h, G^{\text{orig}}, G^{\text{diff}}, Z^h\right)$, the lemma is proved. $\qquad\square$

Next, we demonstrate Theorem 3.2 based on Lemma B.1. According to (Pearl, 2009), the total effect $\text{TE}_{\boldsymbol{z}^f, \boldsymbol{z}^{f'}}(Y^f)$ can be represented as:

$$\begin{aligned}
\text{TE}_{\boldsymbol{z}^f, \boldsymbol{z}^{f'}}(Y^f) &= \text{DE}_{\boldsymbol{z}^f, \boldsymbol{z}^{f'}}(Y^f) + \text{IE}_{\boldsymbol{z}^f, \boldsymbol{z}^{f'}}(Y^f) \\
&= \mathbb{E}\left(Y^f\left(\boldsymbol{z}^{f'}, \{\text{Pa}(Y^f) \setminus Z^f\}(\boldsymbol{z}^f)\right) - \mathbb{E}\left(Y^f(\boldsymbol{z}^f)\right)\right) \\
&\quad + \mathbb{E}\left(Y^f\left(\boldsymbol{z}^f, \{\text{Pa}(Y^f) \setminus Z^f\}(\boldsymbol{z}^{f'})\right) - \mathbb{E}\left(Y^f(\boldsymbol{z}^f)\right)\right),
\end{aligned} \tag{18}$$

where $\text{DE}_{Z,Z'}(Y^f)$ and $\text{IE}_{Z,Z'}(Y^f)$ denotes the natural direct effect and natural indirect effect respectively (Pearl, 2009), $\text{Pa}(\cdot)$ denotes the ancestor variables. $Y^f(\boldsymbol{z}^f)$ denotes the value of $Y^f$ given $Z^f = \boldsymbol{z}^f$ and $Y^f\left(\boldsymbol{z}^{f'}, \{\text{Pa}(Y^f) \setminus Z^f\}(\boldsymbol{z}^f)\right)$ denotes the value of $Y^f$ given $Z^f = \boldsymbol{z}^{f'}$ while parents of $Y^f$ except for $Z^f$ are set as given $Z^f = \boldsymbol{z}^f$.

Then, according to the proof of Lemma B.1, we have:

$$\mathbb{E}\left(Y^f(\boldsymbol{z}^f)\right) = \frac{1}{\mathcal{G}^2} \sum_{G^{\text{orig}} \in \mathcal{G}} f^{\text{latter}}\left(f^{\text{pre}}(G^{\text{orig}}) \setminus f^{Z^f}(G^{\text{orig}}) \cup \boldsymbol{z}^f\right) \tag{19}$$

and, we can also conclude that:

$$Y^f\left(\boldsymbol{z}^{f'}, \{\text{Pa}(Y^f) \setminus Z^f\}(\boldsymbol{z}^f)\right) = f^{\text{latter}}\left(f^{\text{pre}}(G^{\text{orig}}) \setminus f^{Z^f}(G^{\text{orig}}) \cup (\boldsymbol{z}^f \setminus \text{Pa}(Y^f)) \cup (\boldsymbol{z}^{f'} \cap \text{Pa}(Y^f))\right), \tag{20}$$

where $\text{Pa}(Y^f)$ is certain feature variables within model $f(\cdot)$, $G^{\text{orig}}$ represent the input sample here. Likewise, the following equation holds:

$$Y^f\left(\boldsymbol{z}^f, \{\text{Pa}(Y^f) \setminus Z^f\}(\boldsymbol{z}^{f'})\right) = f^{\text{latter}}\left(f^{\text{pre}}(G^{\text{orig}}) \setminus f^{Z^f}(G^{\text{orig}}) \cup (\boldsymbol{z}^{f'} \setminus \text{Pa}(Y^f)) \cup (\boldsymbol{z}^f \cap \text{Pa}(Y^f))\right). \tag{21}$$

Based on equation 19, 20 and 21, we have:

$$
\begin{aligned}
\text{TE}_{\boldsymbol{z}^f, \boldsymbol{z}^{f'}}(Y^f) &= \mathbb{E}\Big(Y^f\big(\boldsymbol{z}^{f'}, \{\text{Pa}(Y^f) \setminus Z^f\}(\boldsymbol{z}^f)\big) - \mathbb{E}\big(Y^f(\boldsymbol{z}^f)\big)\Big) \\
&\quad + \mathbb{E}\Big(Y^f\big(\boldsymbol{z}^f, \{\text{Pa}(Y^f) \setminus Z^f\}(\boldsymbol{z}^{f'})\big) - \mathbb{E}\big(Y^f(\boldsymbol{z}^f)\big)\Big) \\
&= \mathbb{E}\Big(f^{\text{latter}}\big(f^{\text{pre}}(G^{\text{orig}}) \setminus f^{Z^f}(G^{\text{orig}}) \cup (\boldsymbol{z}^f \setminus \text{Pa}(Y^f)) \cup (\boldsymbol{z}^{f'} \cap \text{Pa}(Y^f))\big)\Big) \\
&\quad - \frac{1}{\mathcal{G}^2} \sum_{G^{\text{orig}} \in \mathcal{G}} f^{\text{latter}}\big(f^{\text{pre}}(G^{\text{orig}}) \setminus f^{Z^f}(G^{\text{orig}}) \cup \boldsymbol{z}^f\big)\Big) \\
&\quad + \mathbb{E}\Big(f^{\text{latter}}\big(f^{\text{pre}}(G^{\text{orig}}) \setminus f^{Z^f}(G^{\text{orig}}) \cup (\boldsymbol{z}^{f'} \setminus \text{Pa}(Y^f)) \cup (\boldsymbol{z}^f \cap \text{Pa}(Y^f))\big)\Big) \\
&\quad - \frac{1}{\mathcal{G}^2} \sum_{G^{\text{orig}} \in \mathcal{G}} f^{\text{latter}}\big(f^{\text{pre}}(G^{\text{orig}}) \setminus f^{Z^f}(G^{\text{orig}}) \cup \boldsymbol{z}^f\big)\Big) \\
&= \mathbb{E}\Big(\text{INTINV}(f, (G^{\text{orig}})^{\boldsymbol{z}^f}, (G^{\text{diff}})^{\boldsymbol{z}^{f'}}, Z^f \cap \text{Pa}(Y^f)) \\
&\quad - \frac{1}{\mathcal{G}^2} \sum_{G^{\text{orig}} \in \mathcal{G}} f^{\text{latter}}\big(f^{\text{pre}}(G^{\text{orig}}) \setminus f^{Z^f}(G^{\text{orig}}) \cup \boldsymbol{z}^f\big)\Big) \\
&\quad + \mathbb{E}\Big(\text{INTINV}(f, (G^{\text{orig}})^{\boldsymbol{z}^{f'}}, (G^{\text{diff}})^{\boldsymbol{z}^f}, Z^f \setminus \text{Pa}(Y^f)) \\
&\quad - \frac{1}{\mathcal{G}^2} \sum_{G^{\text{orig}} \in \mathcal{G}} f^{\text{latter}}\big(f^{\text{pre}}(G^{\text{orig}}) \setminus f^{Z^f}(G^{\text{orig}}) \cup \boldsymbol{z}^f\big)\Big),
\end{aligned}
\tag{22}
$$

where $(G^{\text{orig}})^{\boldsymbol{z}^f}$ and $(G^{\text{orig}})^{\boldsymbol{z}^{f'}}$ denotes the graph samples that satisfied $f^{Z^f}((G^{\text{orig}})^{\boldsymbol{z}^f}) = \boldsymbol{z}^f$ and $f^{Z^f}((G^{\text{orig}})^{\boldsymbol{z}^{f'}}) = \boldsymbol{z}^{f'}$. Based on the above derivation and Equation 17, we have:

$$
\begin{aligned}
\text{TE}_{\boldsymbol{z}^f, \boldsymbol{z}^{f'}}(Y^f) &= \mathbb{E}\Big(\text{INTINV}(h, (G^{\text{orig}})^{\boldsymbol{z}^f}, (G^{\text{diff}})^{\boldsymbol{z}^{f'}}, Z^h \cap \text{Pa}(Y^h)) \\
&\quad - \frac{1}{\mathcal{G}^2} \sum_{G^{\text{orig}} \in \mathcal{G}} h^{\text{latter}}\big(h^{\text{pre}}(G^{\text{orig}}) \setminus h^{Z^h}(G^{\text{orig}}) \cup h^{Z^h}((G^{\text{orig}})^{\boldsymbol{z}^f})\big)\Big) \\
&\quad + \mathbb{E}\Big(\text{INTINV}(h, (G^{\text{orig}})^{\boldsymbol{z}^{f'}}, (G^{\text{diff}})^{\boldsymbol{z}^f}, Z^h \setminus \text{Pa}(Y^h)) \\
&\quad - \frac{1}{\mathcal{G}^2} \sum_{G^{\text{orig}} \in \mathcal{G}} h^{\text{latter}}\big(h^{\text{pre}}(G^{\text{orig}}) \setminus h^{Z^h}(G^{\text{orig}}) \cup h^{Z^h}((G^{\text{orig}})^{\boldsymbol{z}^f})\big)\Big) \\
&= \mathbb{E}\Big(Y^h\big(h^{Z^h}((G^{\text{orig}})^{\boldsymbol{z}^{f'}}, \{\text{Pa}(Y^f) \setminus Z^f\}(h^{Z^h}((G^{\text{orig}})^{\boldsymbol{z}^f}))\big) \\
&\quad - \mathbb{E}\big(Y^h(h^{Z^h}((G^{\text{orig}})^{\boldsymbol{z}^f}))\big)\Big) \\
&\quad + \mathbb{E}\Big(Y^h\big(h^{Z^h}((G^{\text{orig}})^{\boldsymbol{z}^f}, \{\text{Pa}(Y^f) \setminus Z^f\}(h^{Z^h}((G^{\text{orig}})^{\boldsymbol{z}^{f'}}))\big) \\
&\quad - \mathbb{E}\big(Y^h(h^{Z^h}((G^{\text{orig}})^{\boldsymbol{z}^f}))\big)\Big) \\
&= \text{TE}_{h^{Z^h}((G^{\text{orig}})^{\boldsymbol{z}^f}), h^{Z^h}((G^{\text{orig}})^{\boldsymbol{z}^{f'}})}(Y^h),
\end{aligned}
\tag{23}
$$

where $\eta^{-1}(\boldsymbol{z}^f)$ can be represented as $h^{Z^h}((G^{\text{orig}})^{\boldsymbol{z}^f})$, the theorem is proved.

## B.2. Proof of Corollary 3.3.

**Corollary 3.3.** *Given the condition that there exist a subset $Z^f$ of the intermediate variables of $f(\cdot)$ satisfied $\eta : Z^f \to Z^h$ where $\eta$ is bijective does not hold, then if $\text{INTINV}\big(f, G^{\text{orig}}, G^{\text{diff}}, Z^f\big) = \text{INTINV}\big(h, G^{\text{orig}}, G^{\text{diff}}, Z^h\big)$ holds, the conclusion given in Theorem 3.2 remains valid.*

As $Z^f = \eta(Z^h)$ no longer holds, we can then locate set $\dot{Z}^f$ that satisfied:

$$\dot{Z}^f = \eta(Z^h \cup U), \tag{24}$$

where $U$ is the minimum set of extra variables that are required to determine the value of $\dot{Z}^f$. We will next analyze the two possible scenarios, namely $U \perp\!\!\!\perp Y^f$ and $U \not\!\perp\!\!\!\perp Y^f$. For case that $U \perp\!\!\!\perp Y^f$, we could fix $U$ as a constant set, and then have the following equation hold:

$$\dot{Z}^f = \eta'(Z^h), \tag{25}$$

where $\eta'(\cdot)$ adopt a set of constant to replace $U$. In such case, we could follow the same demonstration given in Theorem 3.2 and prove that:

$$\text{TE}_{\boldsymbol{z}^f, \boldsymbol{z}^{f'}}(Y^f) = TE_{h^{Z^h}\left((G^{\text{orig}})^{\boldsymbol{z}^f}\right), h^{Z^h}\left((G^{\text{orig}})^{\boldsymbol{z}^{f'}}\right)}(Y^h), \tag{26}$$

For case that $U \not\!\perp\!\!\!\perp Y^f$, if $\text{INTINV}\left(f, G^{\text{orig}}, G^{\text{diff}}, Z^f\right) = \text{INTINV}\left(h, G^{\text{orig}}, G^{\text{diff}}, Z^h\right)$ holds, we could follow Equation 22 and Equation 23 to proof that Equation 26 holds. The corollary is proven.

### B.3. Justification of Proposition 3.4

**Proposition 3.4.** *For a high-level causal model $h(\cdot)$ and a low-level neural network $f(\cdot)$, both mapping input graphs $G \in \mathcal{G}$ to outputs $Y \in \mathcal{Y}$ such that $Y = f(G) = h(G)$, if $Z^f$ within $f(\cdot)$ and $Z^h$ within $h(\cdot)$ minimize $\mathcal{L}_{II}$ to its optimal value $\mathcal{L}_{II}^*$, then $Z^f$ aligns best with $Z^h$ and has an identical total effect on the output prediction as $Z_h$. The minimal $\mathcal{L}_{II}^*$ is given by:*

$$\mathcal{L}_{II}^* = \frac{1}{\mathcal{G}^2} \sum_{G^{orig} \in \mathcal{G}} \sum_{G^{diff} \in \mathcal{G}} \mathcal{D}\left(INTINV(h, G^{orig}, G^{diff}, Z^h), INTINV(h, G^{orig}, G^{diff}, Z^h)\right). \tag{27}$$

The loss $\mathcal{L}_{II}$ defined in Equation 1 can be used to measure the correspondence between the neural network model $f(\cdot)$ and the higher-order causal model $h(\cdot)$.

Specifically, given a high-level causal model $h(\cdot)$ and a low-level neural network model $f(\cdot)$, both of which accurately map input graphs $G \in \mathcal{G}$ to outputs $Y \in \mathcal{Y}$, such that $Y = f(G) = h(G)$, the following holds:

1. For $Z_f$ within $f(\cdot)$ and $Z_h$ within $h(\cdot)$ that let $\mathcal{L}_{II}$ reach the minimal possible value $\mathcal{L}_{II}^*$, then $Z_f$ is the variable that best align with $Z_h$ and holds identical total effect for the output prediction with $Z_h$. $\mathcal{L}_{II}^*$ can be calculated with:

$$\mathcal{L}_{II}^* = \sum_{G^{\text{orig}} \in \mathcal{G}} \sum_{G^{\text{diff}} \in \mathcal{G}} \mathcal{D}\left(\text{INTINV}(h, G^{\text{orig}}, G^{\text{diff}}, Z^h), \text{INTINV}(h, G^{\text{orig}}, G^{\text{diff}}, Z^h)\right). \tag{28}$$

2. For different variables $Z^{f,a}$ and $Z^{f,b}$ within $f(\cdot)$ and the corresponding loss $\mathcal{L}_{II}^a$ and $\mathcal{L}_{II}^b$, if $\mathcal{L}_{II}^a > \mathcal{L}_{II}^b$, we have total effect of $Z^{f,b}$ upon the output $Y$ is strictly similar to $Z^h$ than $Z^{f,a}$ under $D()$, formally:, formally:

$$\text{TE}_{\boldsymbol{z}^f, \boldsymbol{z}^{f'}}(Y^f) - \text{TE}_{\boldsymbol{z}^f, \boldsymbol{z}^{f'}}(Y^f) > \text{TE}_{\boldsymbol{z}^h, \boldsymbol{z}^{h'}}(Y^h). \tag{29}$$

From the proof of Theorem 3.2, we have that the minimal possible value of $\mathcal{L}_{II}$ is that:

$$\mathcal{L}_{II} = \sum_{G^{\text{orig}} \in \mathcal{G}} \sum_{G^{\text{diff}} \in \mathcal{G}} \mathcal{D}\left(\text{INTINV}(h, G^{\text{orig}}, G^{\text{diff}}, Z^h), \text{INTINV}(h, G^{\text{orig}}, G^{\text{diff}}, Z^h)\right), \tag{30}$$

under which Equation 10 holds. According to Corollary 3.3, if Equation 10 holds, and $Y = f(G) = h(G)$, we have the total effect $\text{TE}_{\boldsymbol{z}^f, \boldsymbol{z}'^f}(Y^f)$ of $f(\cdot)$ is equal to the total effect $\text{TE}_{\boldsymbol{z}^h, \boldsymbol{z}'^h}(Y^h)$ of $h(\cdot)$. The conclusion is justified.

### B.4. Proof of Theorem 3.5

**Theorem 3.5.** Given high-level causal model $h(\cdot)$, variable $\bar{Z}^h$ within $h(\cdot)$, if $\bar{Z}^f$ let $\text{INTINV}\left(f, G^{\text{orig}}, G^{\text{diff}}, \bar{Z}^f\right) = \text{INTINV}\left(h, G^{\text{orig}}, G^{\text{diff}}, \bar{Z}^h\right)$, and $G^{\text{orig}}$ and $G^{\text{orig}}$ different in node level, variables within the GNN model are sufficient to align $\bar{Z}^f$.

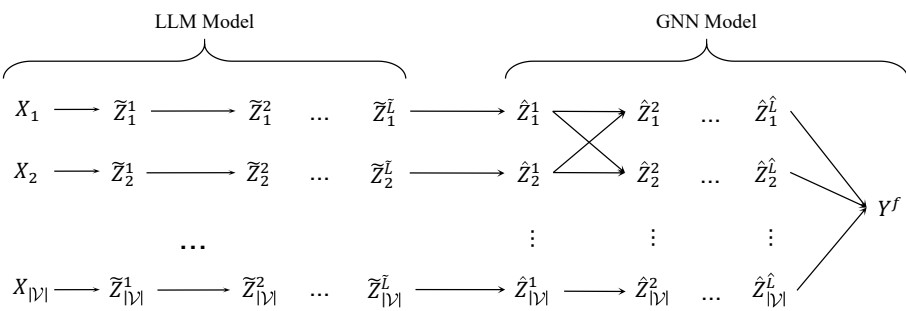

*Figure 8.* The graphical representation of the proposed SCM. Elements in set $\{\widetilde{Z}_j^{i,\widetilde{l}-1}\}_{j=1}^{\widetilde{H}_{\widetilde{l}}}$ holds right-arrows point to $\widetilde{Z}_j^{i,\widetilde{l}}$ and $\{\hat{Z}_k^{\hat{l}-1}|k \in \mathcal{N}(i)\cup\{i\}\}$ holds right-arrows point to $\hat{Z}_i^{\hat{l}}$. Some of the above-mentioned arrows are omitted due to complexity. The red arrows represent relationships that may not necessarily exist.

To prove Theorem 3.5, we need to conduct a more detailed and accurate analysis of the variables within the model. Therefore, we introduce the concept of SCM (Structural Causal Model) (Pearl, 2009), which is a framework that uses equations and directed graphs to represent and analyze causal relationships between variables.

Figure 8 shows the constructed SCM. In the figure, some variables have been omitted, and edges connecting or crossing the ellipses have also been removed from the demonstration, e.g., edge $\hat{Z}_1^1 \to \hat{Z}_{|\mathcal{V}|}^2$ is removed as it across the ellipses. In Figure 8, we use $X_i$ to denote the node attribute of $i$-th node, $\widetilde{Z}_j^{\widetilde{l}}$ represents the output variable of the $\widetilde{l}$-th hidden layer of the LLM according to the $j$-th node feature. $\hat{Z}_i^{\hat{l}}$ represents the node representation of the $i$-th node of layer $\hat{l}$ in the GNN.

Within the SCM, for $\hat{Z}_i^{\hat{l}}$, we have set $\mathrm{Pa}(\hat{Z}_i^{\hat{l}}) = \{\hat{Z}_k^{\hat{l}-1}|k \in \mathcal{N}(i) \cup \{i\}\}$, where $\mathcal{N}(i)$ denotes the neighboring nodes of node $i$. $\mathrm{Pa}(\hat{Z}_i^{\hat{l}}) = \{\hat{Z}_k^{\hat{l}-1}|k \in \mathcal{N}(i) \cup \{i\}\}$ implies that $\mathrm{Pa}(\hat{Z}_i^{\hat{l}})$ consists of the node representations of the last layer that the corresponding node is adjacent to or the same as node $i$. We support the validity of the proposed SCM through the following lemma and its corresponding proof.

**Lemma B.2.** *The SCM that is demonstrated within figure 8 can represent the general causal relationships among variables in the LLM-enhancer-plus-GNN model.*

*Proof.* We can demonstrate Lemma B.2 by showing that such SCM is consistent with the results generated using the IC algorithm (Pearl, 2009) for SCM construction. The IC algorithm is a commonly used method for constructing SCMs, please refer to section 3.2 of (Pearl, 2009) for details. We follow the triple-step method of the IC algorithm for the proof.

**Step 1.** We connect $X_i$ with $\widetilde{Z}_i^1$, as there exist none set $G^{\mathrm{diff}}$ that satisfied $X_i \perp\!\!\!\perp \widetilde{Z}_i^1|G^{\mathrm{diff}}$. Furthermore, we connect $\widetilde{Z}_i^{\widetilde{l}}$ with $\widetilde{Z}_i^{\widetilde{l}-1}$, due to the neural network architecture used by the LLM. We no longer connect any other variables in the LLM. Then, $\hat{Z}_j^{\widetilde{L}}$ is connected to $\widetilde{Z}_j^1$, as such elements form the node representation of the GNN. Next, $\{\hat{Z}_k^{\hat{l}-1}|k \in \mathcal{N}(i) \cup \{i\}\}$ and $\hat{Z}_i^{\hat{l}}$ is connected, as the node representation relies on the node representations of last layer that the corresponding node is adjacent to or the same as node $i$. $Y^f$ and $\{\hat{Z}_k^{\hat{L}}\}_{k=1}^{|\mathcal{V}|}$ are connected as $Y^f$ calculated through global pooling. For node classification, the predicted label shall connect with the elements within $\{\hat{Z}_k^{\hat{L}}\}_{k=1}^{|\mathcal{V}|}$ correspondingly. And the others are not connected for the same reason as the LLM part.

**Step 2.** Based on the feed-forward mechanism of neural networks, all edges should be oriented in the feed-forward direction.

**Step 3.** Since all edges are already directed, step three is unnecessary.

The constructed SCM is consistent with that in Figure 8, thus the lemma is proven. □

With the proposed SCM within Figure 8 and Lemma B.2, we carry on demonstrating the proposed theorem. As the SCM

demonstrates, we have $\hat{Z}_j^1$ block all causal routes from $X_j$ to $Y^f$. Therefore, we have:

$$\text{INTINV}\big(f, G^{\text{orig}}, G^{\text{diff}}, S\big) = \text{INTINV}\big(h, G^{\text{orig}}, G^{\text{diff}}, \hat{Z}_j^1\big), S \in \{X_j\} \cup \{\widetilde{Z}_j^i\}_{i=1}^{\widetilde{L}}. \tag{31}$$

If $\forall S \in \bar{Z}^f, S \notin \left\{ \widetilde{Z}_j^i \mid 1 \le j \le |\mathcal{V}|, 1 \le i \le \widetilde{L} \right\}$, then the conclusion given in the theorem naturally holds. Otherwise, for case that $\exists S \in \bar{Z}^f, S \in \left\{ \widetilde{Z}_j^i \mid 1 \le j \le |\mathcal{V}|, 1 \le i \le \widetilde{L} \right\}$ we have:

$$\text{INTINV}\big(f, G^{\text{orig}}, G^{\text{diff}}, \bar{Z}^f\big) = \text{INTINV}\big(f, G^{\text{orig}}, G^{\text{diff}}, \widetilde{S} \cup \hat{S}\big)$$
$$= \text{INTINV}\big(f, (G^{\text{orig}})^{\hat{S}}, G^{\text{diff}}, \widetilde{S}\big) \tag{32}$$

where $\widetilde{S} \subseteq \left\{ \widetilde{Z}_j^i \mid 1 \le j \le |\mathcal{V}|, 1 \le i \le \widetilde{L} \right\}, \hat{S} \subseteq \left\{ \hat{Z}_j^i \mid 1 \le j \le |\mathcal{V}|, 1 \le i \le \hat{L} \right\}, (G^{\text{orig}})^{\hat{S}}$ denotes the input that ensure $\hat{S}$ values changes to that of $G^{diff}$ as input. Based on equation 31, we have:

$$\text{INTINV}\big(f, (G^{\text{orig}})^{\hat{S}}, G^{\text{diff}}, \widetilde{S}\big) = \text{INTINV}\big(f, (G^{\text{orig}})^{\hat{S}}, G^{\text{diff}}, \hat{S}'\big),$$
$$\hat{S}' \subseteq \left\{ \hat{Z}_j^i \mid 1 \le j \le |\mathcal{V}|, 1 \le i \le \hat{L} \right\}. \tag{33}$$

Therefore, we have the following holds for certain $\mathcal{S}$:

$$\text{INTINV}\big(f, G^{\text{orig}}, G^{\text{diff}}, \bar{Z}^f\big) = \text{INTINV}\big(f, G^{\text{orig}}, G^{\text{diff}}, \mathcal{S}\big) \tag{34}$$
$$= \text{INTINV}\big(h, G^{\text{orig}}, G^{\text{diff}}, \bar{Z}^h\big), \tag{35}$$
$$\mathcal{S} \subseteq \left\{ \hat{Z}_j^i \mid 1 \le j \le |\mathcal{V}|, 1 \le i \le \hat{L} \right\}. \tag{36}$$

The theorem is proved.

## C. Details of The CCSG Dataset

### C.1. Node Features

Below, we provide a detailed introduction to the node features in CCSG. All data related to CCSG, as well as the methods used to construct the data, will be made open-source. *We have also provided the corresponding data and code in the supplementary materials.*

### C.1.1. SELF-CONSTRUCTED FEATURES

We independently constructed a portion of the node features with the primary objective of thoroughly analyzing how graph representation learning models effectively handle graph data when the node features are relatively simple. This design aims to ensure that the information within the dataset is predominantly reflected in the graph's topological structure, minimizing the interference of complex node features with model performance. By doing so, we can better investigate the model's reliance on and ability to process structural information.

Specifically, we adopted two feature construction methods: First, we generated noise features based on random distributions. This approach introduces randomness to ensure feature neutrality while eliminating the influence of specific patterns, thereby enabling us to observe the model's ability to capture topological information under random conditions. Second, we directly assign feature values to nodes based on their categories, such as assigning unique feature identifiers to nodes of each class. This provides the model with the most basic classification information and allows us to test its performance under minimal feature conditions.

By employing these two feature construction methods, we ensure that the node feature design meets the analytical requirements while functioning as a controlled variable. This not only enhances the precision and interpretability of our experiments but also establishes a solid foundation for exploring the central role of graph structural information in model learning.

We extracted a large number of entries from Wikipedia to construct semantically rich node features with known interrelations. As illustrated in Table 3, the entries we collected are categorized into three main classes: spaceflight, computer, and software. These domains are further subdivided into 15 specific subcategories, providing a fine-grained classification structure. Additionally, we recorded interlinking information among the entries in the dataset to facilitate the construction of training data with explicit causal relationships. Table 4 presents three representative examples of the data we collected, demonstrating the diversity and structure of the dataset. The contents of the "Background Categories," "Related Terms," and "Similar Entries" fields will be used to identify similar nodes for constructing a coherent graph structure. This approach ensures a comprehensive and well-documented framework for downstream applications and analysis.

*Table 3.* Classes of the entries collected from Wikipedia.

| Class | Count |
|---|---|
| **Spaceflight** | **1665** |
| Spaceflight | 401 |
| Satellite | 401 |
| Rocket | 260 |
| Outer Space | 302 |
| Space Science | 301 |
| **Computer** | **2205** |
| Computer engineering | 401 |
| Automation | 401 |
| Computer security | 601 |
| Computer science | 401 |
| Computer hardware | 401 |
| **Software** | **2005** |
| Computer programming | 401 |
| Software testing | 401 |
| Application software | 401 |
| Software development | 401 |
| Software architecture | 401 |

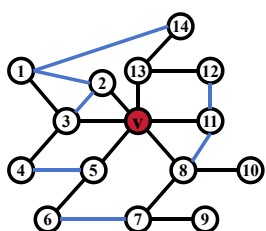

*Figure 9.* Example Graph.

## C.2. Node Correlations

With a diverse set of rich, multi-category node features in place, the next step in constructing a graph dataset is to build its topological structure. One approach to this is to base the construction on the interrelationships between nodes, connecting them through associations and simple categorical relationships. For instance, this method is partially employed in the analytical experiments presented in the main text. In this work, we leverage the interrelationships extracted from Wikipedia entries, combined with manually crafted node features, to define these connections. This ensures that the causal relationships between nodes are explicitly established, resulting in a well-defined and interpretable graph topology.

## C.3. Topological Structures

We constructed a total of 10 distinct topological structures, with their types and specific configurations detailed in Table 5. In addition to commonly used graph structures in causal representation learning analyses (Wu et al., 2022; Gao et al., 2024), we incorporated a variety of graph topologies from graph theory, such as complete graphs and bipartite graphs. This was done to ensure comprehensive coverage of potential graph structures that could have causal effects on classification tasks. By diversifying the graph topologies, our approach aims to provide a robust framework for evaluating the influence of graph structure on causal representation learning and downstream analytical performance.

*Table 4.* Samples of the collected data

| Name | Content | Class | SubClass | Background Categories | Related Terms | Similar Entries |
|---|---|---|---|---|---|---|
| Computational social choice | Computational social choice is the study of problems that arise from aggregating the preferences of a group of agents using computer science and social choice theory. It focuses on efficiently computing voting outcomes, the complexity of manipulation, and representing preferences in combinatorial contexts. | Computer | Computer engineering | Social choice theory, Voting theory, Computer science | Voting theory, Social choice theory, Egalitarian rule, Utilitarian rule, Agreeable subset, Anonymity, Arrow's impossibility theorem, Bayesian regret, Budget-proposal aggregation, Computational social choice, Dictatorship mechanism, Discursive dilemma, Electoral list, Electoral system, Extended sympathy, Fractional approval voting, Fractional social choice, Gibbard–Satterthwaite theorem, Gibbard's theorem, Implicit utilitarian voting, Independence of irrelevant alternatives, Intensity of preference, Liberal paradox, Christian List, May's theorem, McKelvey–Schofield chaos theorem, Mechanism design, Median graph, Median voting rule, Nakamura number, Neutrality, Optimal apportionment, Proportional-fair rule, Quasitransitive relation, Ranked voting, Rated voting, Sequential elimination method, Social Choice and Individual Values, Social welfare function, Unrestricted domain | Algocracy, Algorithmic game theory, Algorithmic mechanism design, Cake-cutting, Fair division, Hedonic games |
| CEN/XFS | CEN/XFS (extensions for financial services) is a client-server architecture for financial applications on the Microsoft Windows platform, especially for devices like EFTPOS terminals and ATMs. It is an international standard promoted by the European Committee for Standardization and is based on the WOSA Extensions for Financial Services developed by Microsoft. XFS allows financial institutions to choose t | Computer | Computer engineering | Windows communication and services, Device drivers, Embedded systems | .NET Remoting, Administrative share, Bonjour Sleep Proxy, CEN/XFS, Channel Definition Format, Discovery and Launch, Distributed Component Object Model, Dynamic Data Exchange, EternalBlue, Indexing Service, Internet Connection Sharing, Internet Locator Server, Ipconfig, Layered Service Provider, Link Layer Topology Discovery, Link-Local Multicast Name Resolution, List of products that support SMB, LMHOSTS, Local Inter-Process Communication, Microsoft Message Passing Interface, Microsoft Message Queuing, Poison message, Microsoft Messenger service, Microsoft Transaction Server | Xpeak, Automated teller machine, Teller assist unit |
| iDempiere | iDempiere is an open source ERP software fully navigable on various devices. It includes CRM and SCM functions, and is community powered, unlike proprietary ERP solutions. | Software | Software Development | Free ERP software, Free business software | Adaxa Suite, Adempiere, Apache OFBiz, Compiere, Dolibarr, ERP5, ERPNext, HeliumV, IDempiere, InoERP, IntarS, JFire, Kuali Foundation, LedgerSMB, Metasfresh, Odoo, Postbooks, Tryton | OSGI, Java, Compiere, List of ERP software packages |
| Kallithea (software) | Kallithea is a free and cross-platform source code management system that focuses on providing repository hosting services for collaboration. It offers features such as forking, pull requests, code review, and issue tracking. Kallithea was created as a fork of RhodeCode due to changes in the license terms. While earlier versions of RhodeCode were licensed under the GNU GPL version 3, version 2.0 introduced exceptions | Software | Software Development | Open-source hosted development tools, Project management software, Free software programmed in Python, Free project management software | Cloud9 IDE, GitLab, Gitorious, Kallithea (software), Travis CI | Comparison of project management software, List of tools for code review, Comparison of source code hosting facilities, Apache Allura |
| Lunar Reconnaissance Orbiter #Payload | The Lunar Reconnaissance Orbiter (LRO) is a NASA spacecraft currently orbiting the Moon, collecting essential data for future human and robotic missions. Launched in 2009 as part of the Lunar Precursor Robotic Program, LRO has created a detailed 3-D map of the lunar surface at 100-meter resolution, including high-resolution images of Apollo landing sites | Spaceflight | Space Science | Lunar Reconnaissance Orbiter, Missions to the Moon, NASA space probes, Space probes launched in 2009, Satellites orbiting the Moon | Lunar Reconnaissance Orbiter, Diviner, LCROSS, Mini-RF | Exploration of the Moon, LCROSS, List of missions to the Moon, Lunar Atmosphere and Dust Environment Explorer, Lunar outpost (NASA), Lunar water |

*Table 5.* Details concerning the utilized topological structures.

| Structure Type | Grid | Circle | Chain | Tree | Star |
|---|---|---|---|---|---|
| **Illustration** |  |  |  |  |  |
| **Description** | Nodes are connected in a grid pattern, and the number of nodes is adjustable. | Nodes form a closed loop, and the number of nodes is adjustable. | Nodes are connected in a chain without branches, useful for over-squashing. | Nodes form a tree, branching and height are adjustable. | Multiple nodes connect to a central node, and the number of nodes is adjustable. |
| **Structure Type** | Complete Graph | Chordal Cycle | Bipartite Graph | Bipartite Graph | Fixed Structure |
| **Illustration** |  |  |  |  |  |
| **Description** | Every pair of nodes is connected, and the number of nodes is adjustable. | Graph with chords in the cycle, with adjustable nodes and chords. | Vertices are divided into two disjoint sets, adjustable nodes. | Outer nodes connected in a ring with a central node, adjustable nodes. | A random graph structure with fixed topology. |

*Table 6.* Accuracy corresponding to different $q$ values, using different numbers of prompts. Llama3 was used as the enhancer. The standard deviation of results over 10 trials.

| Dataset | $q = 1$ | $q = 2$ | $q = 3$ | $q = 4$ |
|---|---|---|---|---|
| Cora | 84.94±0.81 | 86.45±0.49 | 85.79±0.51 | 83.33±0.23 |
| Pubmed | 83.84±0.77 | 84.32±1.06 | 83.98±0.39 | 82.58±0.44 |

## C.4. Node-level Analytical Experiments Dataset Details

For node-level classification datasets, as described in Section 3.3.1, we fix the nodes to be classified and ensure that their associated causal relationships are controllable and analyzable. A specific example is provided in Figure 9. We further utilize the subcategories and interrelationship information included in CCSG to construct edges between the related nodes. At the same time, we partition the nodes into sets based on the shortest path lengths from each node to the target node $v$ and the associations between nodes. The attributes of each node set are processed by $h^{\text{node},1}$. The outputs of $h^{\text{node},1}$ serve as the inputs to $h^{\text{node},2}(\cdot)$. The function determines the output based on the categories of the node itself and its neighboring nodes. These outputs will serve as the input for $h^{\text{node},3}(\cdot)$, which produces the final classification result for the node. $h^{\text{node},3}(\cdot)$ takes as input the outputs of $h^{\text{node},2}(\cdot)$ corresponding to the multi-order neighbors of the target node $v$ and generates the final result.

Specifically corresponding to the experimental results in Figures 1 and 2, $\Psi_1$, $\Psi_2$, and $\Psi_3$ represent the categories of nodes at distances of 1, 2, and 3 from node $v$, respectively. These categories are determined by the predominant category of these nodes. Subsequently, $\Phi_1$ and $\Phi_2$ are features constructed based on the categories of nodes and the categories of their neighboring nodes. The difference lies in that $\Phi_2$ considers multi-order neighboring nodes.

For node-level analytical experiments, we construct up to 3,000 graph samples, each containing a single target node for classification, 3 classes. On average, each graph consists of 28.9 nodes. We select 500 samples for testing and all samples for training. The test set is utilized for interchange intervention-based analysis. To reduce computational complexity, we also exclude sample pairs that remain completely unchanged before and after the interchange intervention based on $h(\cdot)$.

## C.5. Graph-level Analytical Experiments Dataset Details

For graph-level classification datasets, we utilize the topological structures and node features together. Specifically, we use certain substructures and their connections within the topological structures as the output of $h^{\text{graph},1}(\cdot)$. Subsequently, the output of $h^{\text{graph},2}(\cdot)$ represents the category of the entire topological structure, while $h^{\text{graph},3}(\cdot)$ produces the final result based on the category of the topological structure and its connection patterns.

In the results shown in Figures 1 and 2, $\Gamma_1$, $\Gamma_2$, and $\Gamma_3$ are defined as substructures of the topological structures, differing in the number of nodes they contain: $\Gamma_1$ consists of structures with 3 nodes, $\Gamma_2$ consists of structures with 6 nodes, and $\Gamma_3$ consists of structures with 9 nodes. $\Omega_1$ and $\Omega_2$ specifically refer to the types of graph structures commonly used in causal representation learning analyses, as well as the graph topologies derived from graph theory. Both types of structures are present in each sample.

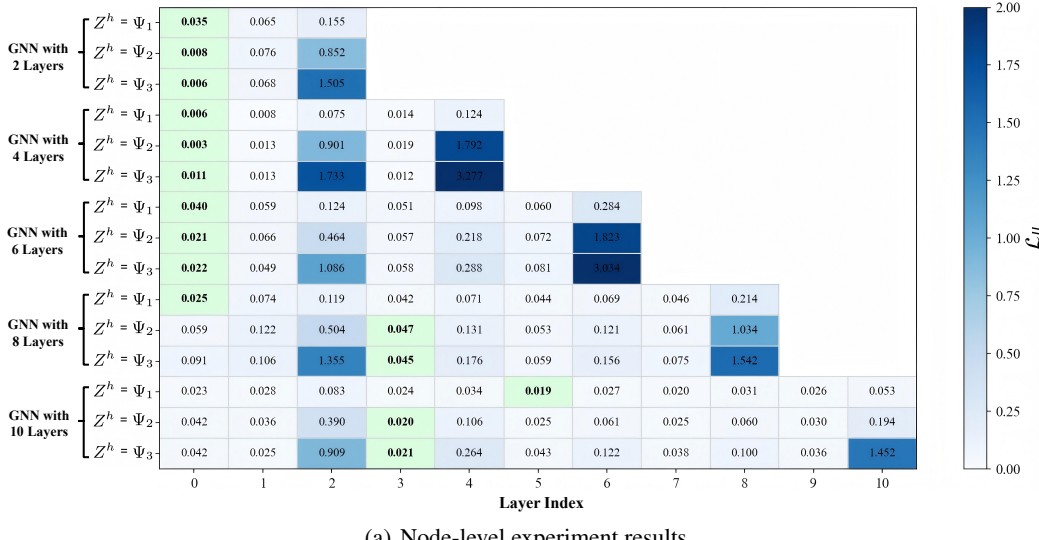

(a) Node-level experiment results.

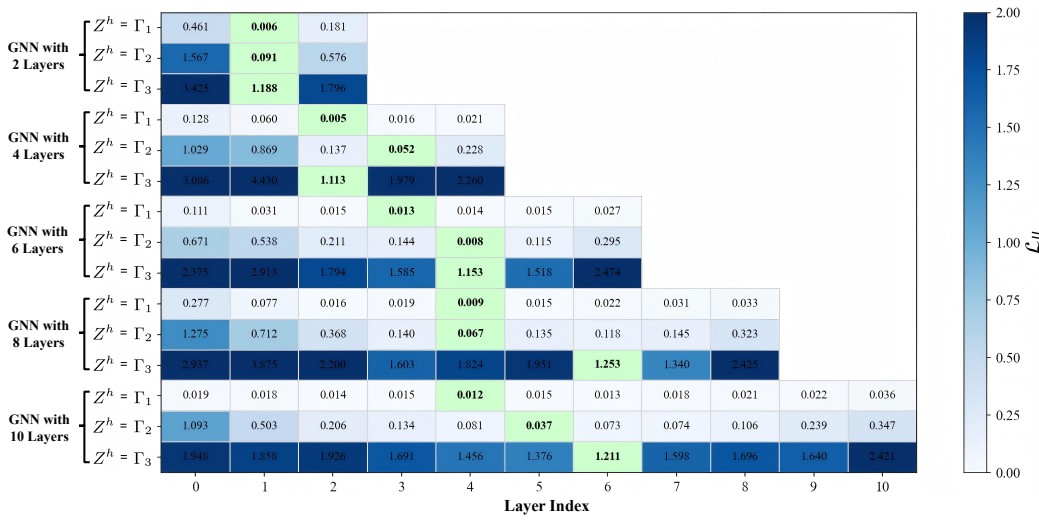

(b) Graph-level experiment results.

*Figure 10.* Alignment results of graph-level and node-level experiments with varying GNN layers.

For graph-level analytical experiments, we construct up to 1500 graph samples, use a total of 10 topological structures, each of which consists of 20.5 node. We select 300 samples for testing and all samples for training. We utilize these samples in Node-level Analytical Experiments.

## D. Experimental Details

### D.1. Hyperparameters

We conducted all our experiments on a computer equipped with a single A100 GPU, running the Ubuntu 20.04 operating system. The learning rate for the model was set to 0.001, and the total number of training epochs was 200. For the LLM components, we utilized pre-trained parameters from publicly available open-source models, keeping them fixed during our experiments. Unless explicitly stated otherwise, Llama3 and a GCN with 4 layers and 256 hidden dimensions (if not specifically declared) were employed as the backbone networks in all analytical experiments. For experiments related to the AT module, the hyperparameter $m$ was set to 2, $\delta$ is set to 10, and $q$ was set to 10.

## D.2. Implementation Details of the AT module

The AT module utilizes a Transformer with a 4-layer network and a hidden dimension of 512. It employs an LLM to automatically generate prompts, with the content of the prompt as follows:

> **Prompt Generation**
>
> In order to summarize the content of **<data information>**, please provide **<q>** different prompts. The prompts should be in the format of [Prompt Content] + [Content to be Summarized]. Only output the [Prompt Content].

**<data information>** refers to the details regarding the dataset, whereas **<q>** represents the hyperparameter $q$. It is worth noting that the AT module also supports the use of manually designed prompts, which can help in selecting more effective prompts.

# E. Extra Experiments.

In Figures 10 and 11, we provide a more comprehensive presentation of the experimental results, including the alignment outcomes from several additional experiments. It is evident from all the experimental results that the phenomenon described in Empirical Finding 2 consistently emerges—indicating that, as the size of the model changes, it retains certain specific logic structures that remain unaffected by these variations. Furthermore, the results illustrated in Figure 12 also reveal the correlation between the optimal $\mathcal{L}_{\mathrm{II}}$ and the model's accuracy. Figure 13 presents further experiments conducted with different datasets. Finally, Table 6 summarizes the hyperparameter experiment for $q$.

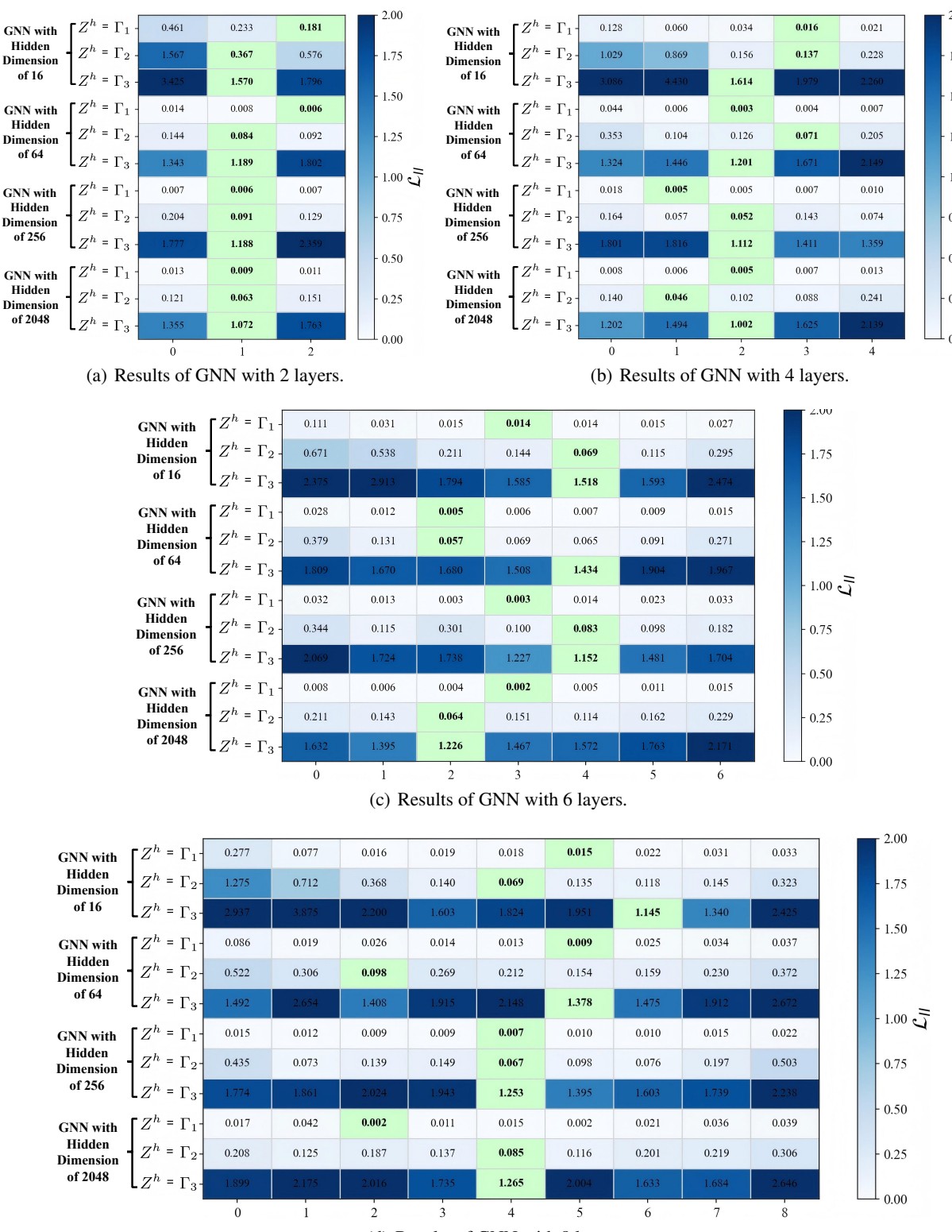

*Figure 11.* Alignment results of graph-level experiments with varying GNN layers and hidden dimensions.

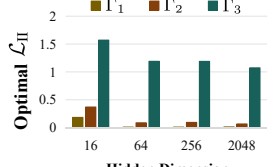 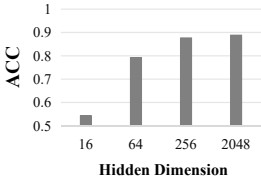 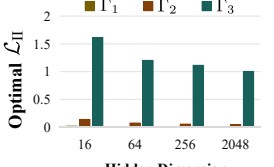 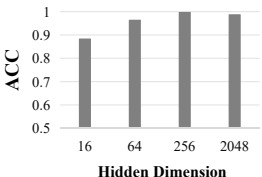

(a) Graph-level task alignment results with the number of GNN layers set to 2 and varying hidden dimensions.

(b) Graph-level task model accuracy with the number of GNN layers set to 2 and varying hidden dimensions.

(c) Graph-level task alignment results with the number of GNN layers set to 4 and varying hidden dimensions.

(d) Graph-level task model accuracy with the number of GNN layers set to 4 and varying hidden dimensions.

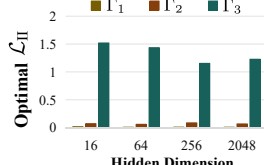 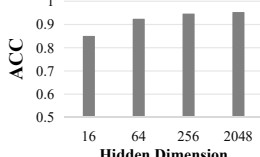 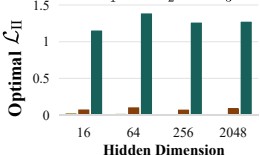 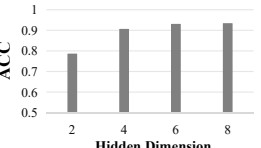

(e) Graph-level task alignment results with the number of GNN layers set to 6 and varying hidden dimensions.

(f) Graph-level task model accuracy with the number of GNN layers set to 6 and varying hidden dimensions.

(g) Graph-level task alignment results with the number of GNN layers set to 8 and varying hidden dimensions.

(h) Graph-level task model accuracy with the number of GNN layers set to 8 and varying hidden dimensions.

*Figure 12.* Experimental results under different number of GNN layers and hidden dimensions.

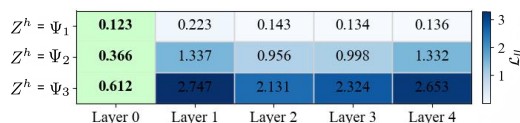
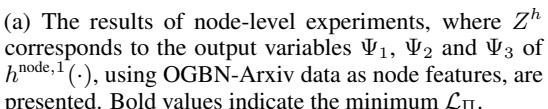
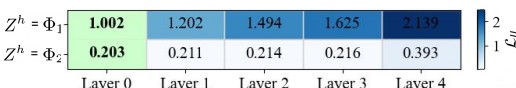

(a) The results of node-level experiments, where $Z^h$ corresponds to the output variables $\Psi_1$, $\Psi_2$ and $\Psi_3$ of $h^{\text{node},1}(\cdot)$, using OGBN-Arxiv data as node features, are presented. Bold values indicate the minimum $\mathcal{L}_\Pi$.

(b) The results of node-level experiments, where $Z^h$ corresponds to the output variables $\Phi_1$ and $\Phi_2$ of $h^{\text{node},2}(\cdot)$, using random data as node features, are presented. Bold values indicate the minimum $\mathcal{L}_\Pi$.

*Figure 13.* Extra experiments using OGBN-Arxiv and random data.

