# OpenReview forum: "LLM Enhancers for GNNs: An Analysis from the Perspective of Causal Mechanism Identification"
_ICML.cc/2025/Conference — ICML 2025 poster_

### Official Review · Reviewer_XFjz · 2025-03-10

**Overall Recommendation:** 4

**Summary:**

This paper explores the use of large language models (LLMs) as feature enhancers for graph neural networks (GNNs) in graph representation learning, addressing the fundamental properties of this approach using the interchange intervention method from causality theory. To facilitate analysis, the authors construct a synthetic graph dataset with controllable causal relationships, enabling precise manipulation of semantic structures. Through systematic interchange interventions, they investigate the correspondence between the LLM-enhancer-plus-GNN model and a high-order causal model, uncovering the internal logical structure of the black-box neural network. Based on these insights, they propose a plug-and-play optimization module to improve information transfer between the LLM enhancer and the GNN. Experimental validation across multiple datasets and models demonstrates the effectiveness of this module, offering a deeper understanding and improved performance of LLM-enhanced GNNs.

**Claims And Evidence:**

The paper provides clear and structured evidence for its claims through a combination of theoretical analysis, synthetic dataset experiments, and empirical validation across multiple models and datasets. The use of a synthetic graph dataset with controllable causal relationships strengthens the credibility of the analysis by allowing precise manipulation of semantic structures and causal dependencies.

**Essential References Not Discussed:**

N/A

**Experimental Designs Or Analyses:**

The proposed analysis framework should be more generalizable, and the Controlled Causal-Semantic Graph dataset should be one of its implementations. Or, in my opinion, the Controlled Causal-Semantic Graph dataset should be used to evaluate the proposed analysis framework.

**Methods And Evaluation Criteria:**

The entire analysis is conducted on a specific dataset, namely the Controlled Causal-Semantic Graph. My concern is the difference between a Causal-Semantic Graph and a causal graph. Based on my understanding of this dataset, the causality is still represented in a semantic way rather than reflecting the actual mechanism.

**Other Comments Or Suggestions:**

Line 251: LM-enhancer-plus-GNN -> LLM-enhancer-plus-GNN

**Other Strengths And Weaknesses:**

The paper presents a novel perspective by applying causal analysis through interchange interventions to investigate the role of LLM-enhanced GNNs. The originality lies in its effort to bridge causality and graph representation learning, particularly in quantifying how LLM-derived features align with causal structures in GNNs. The introduction of the Controlled Causal-Semantic Graph (CCSG) dataset is another notable contribution, as it enables a structured evaluation of causal relationships in graph-based learning.

However, one limitation is the generality of the findings, as the analysis is heavily dependent on the CCSG dataset. While synthetic datasets allow for precise control over causal structures, it remains unclear whether the insights generalize to structured causal DAGs rather than a semantic causal graph.

**Questions For Authors:**

N/A

**Relation To Broader Scientific Literature:**

The key contributions of this paper build upon and extend prior research in understanding how LLM enhancers contribute to GNNs through a causal lens.

**Theoretical Claims:**

Yes

---

> ### Author Rebuttal · Authors · 2025-03-26
>
> Thank you very much for your analysis and suggestions for revising our paper,  and your support for our paper. We greatly appreciate your feedback. Below, we will address each of the raised concerns.
>
> **Methods And Evaluation Criteria**:
> In the appendix, we provide a detailed description of the composition of our dataset. In fact, the semantic information in the Controlled Causal-Semantic Graph dataset is primarily reflected in the semantic richness of its node features. The relationships and labels in the graph are determined based on the entry categories and associations provided by Wikipedia, following a deterministic approach to ensure causality.
>
> **Experimental Designs Or Analyses**：
> Our proposed method can be applied to any graph dataset capable of abstracting higher-order causal models, a point we will clearly mention in the paper. Achieving this currently requires synthetic graph datasets with controllable causal relationships, where node features must contain rich semantic information to demonstrate the effectiveness of LLMs. Due to the limited availability of such datasets, we built our own CCSG dataset. To validate the generalizability of our method, we incorporated node data from the ogbn-arxiv dataset and reconducted the node-level experiments, we also conduct the experiments using random data, with the results as follows:
>
> **Table 1: The results of node-level experiments, where $ Z^{h} $ corresponds to the output variables $ \Psi_1 $,$ \Psi_2 $, and$ \Psi_3 $ of $h^{\text{node,1}}(\cdot) $, using OGBN-Arxiv data and random data as node features, are presented. These results are consistent with those in the paper. Bold values indicate the minimum $ \mathcal{L}_{\text{II}} $.**
>
> |   | Layer 0             | Layer 1             | Layer 2             | Layer 3             | Layer 4             |
> |--------|---------------------|---------------------|---------------------|---------------------|---------------------|
> |$\Psi_1$     | **0.123**      | 0.223      | 0.143  | 0.134      | 0.136      |
> |$\Psi_2$    | **0.366**      | 1.337      | 0.956     | 0.998      | 1.332      |
> |$\Psi_3$     | **0.612**      | 2.747      | 2.131      | 2.324      | 2.653      |
>
> **Table 2: The results of node-level experiments, where $ Z^{h} $ corresponds to the output variables $ \Phi_1 $ and $ \Phi_2 $ of $h^{\text{node,2}}(\cdot) $, using random data as node features, are presented. These results are consistent with those in the paper. Bold values indicate the minimum $ \mathcal{L}_{\text{II}} $.**
>
> |   | Layer 0             | Layer 1             | Layer 2             | Layer 3             | Layer 4             |
> |-----------|-----------|-----------|-----------|-----------|-----------|
> |$\Psi_1$     | 1.002    | **1.202** | 1.494    | 1.625    | 2.139     |
> |$\Psi_1$     | 0.203    | **0.211**| 0.214    | 0.216    | 0.393    |
>
> We will add these results to the paper as well.

---

### Official Review · Reviewer_iycs · 2025-03-12

**Overall Recommendation:** 3

**Summary:**

This paper presents a valuable analysis of the LLM-enhancer-plus-GNN framework, exploring its underlying mechanisms and identifying potential areas for improvement. The use of the CCSG dataset and the interchange intervention method provides a novel approach to understanding the relationship between LLMs and GNNs. The proposed AT module offers a practical solution to enhance the information transfer between these components, leading to improved performance. The paper is well-written and the results are presented clearly.

**Claims And Evidence:**

The claims made in the submission supported by clear and convincing evidence.

**Essential References Not Discussed:**

In my understanding, there are none.

**Experimental Designs Or Analyses:**

Lack the Analysis of hyperparameter q :In Chapter 4, the impact of setting the hyperparameter q on the results lacks sufficient discussion.

**Methods And Evaluation Criteria:**

Overall, the methods and/or evaluation criteria proposed in this paper are reasonable. Here are two additional suggestions.

Suggestions:
1. Extend the Analysis to Other Benchmark: Evaluate the performance of the method on more datasets, such as ognb-arxiv, etc.
2. Compare with Other Methods: Conduct a more comprehensive comparison with other LLM-enhancer-plus-GNN methods.

**Other Comments Or Suggestions:**

None.

**Other Strengths And Weaknesses:**

None.

**Questions For Authors:**

None.

**Relation To Broader Scientific Literature:**

The authors evaluated the impact of modifying the features transmitted from the LLM enhancer to the GNN on model performance, and discovered that token position selection in the LLM enhancer output has a significant effect on model performance. This finding is novel and can be used to improve the performance of the LLM enhancer + GNN framework.

**Theoretical Claims:**

The theoretical Claims proposed by the authors of this paper are relatively reasonable.
In the node-level and graph-level analysis, the authors used the CCSG dataset and causal modeling methods to analyze the llm-augmentor+gnn framework, revealing that for a fixed-parameter LLM enhancer, the features output by the LLM have the function of representing information at both the node level and the raw data level.

---

> ### Author Rebuttal · Authors · 2025-03-31
>
> Thank you very much for your analysis and suggestions for revising our paper, and your support for our paper. We greatly appreciate your feedback. Below, we will address each of the raised concerns.
>
> **Methods And Evaluation Criteria:**
> 1. Thank you for your suggestions. We have incorporated data from the ogbn-arxiv dataset and conducted additional experiments. The results are as follows:
>
> **Table 1: The results of node-level experiments, where $ Z^{h} $ corresponds to the output variables $ \Psi_1 $,$ \Psi_2 $, and$ \Psi_3 $ of $h^{\text{node,1}}(\cdot) $, using OGBN-Arxiv data and random data as node features, are presented. These results are consistent with those in the paper. Bold values indicate the minimum $ \mathcal{L}_{\text{II}} $.**
>
> |   | Layer 0             | Layer 1             | Layer 2             | Layer 3             | Layer 4             |
> |--------|---------------------|---------------------|---------------------|---------------------|---------------------|
> |$\Psi_1$     | **0.123**      | 0.223      | 0.143  | 0.134      | 0.136      |
> |$\Psi_2$    | **0.366**      | 1.337      | 0.956     | 0.998      | 1.332      |
> |$\Psi_3$     | **0.612**      | 2.747      | 2.131      | 2.324      | 2.653      |
>
> We also conducted experiments using random data, and the results are as follows:
>
> **Table 2: The results of node-level experiments, where $ Z^{h} $ corresponds to the output variables $ \Phi_1 $ and $ \Phi_2 $ of $h^{\text{node,2}}(\cdot) $, using random data as node features, are presented. These results are consistent with those in the paper. Bold values indicate the minimum $ \mathcal{L}_{\text{II}} $.**
>
> |   | Layer 0             | Layer 1             | Layer 2             | Layer 3             | Layer 4             |
> |-----------|-----------|-----------|-----------|-----------|-----------|
> |$\Psi_1$     | 1.002    | **1.202** | 1.494    | 1.625    | 2.139     |
> |$\Psi_1$     | 0.203    | **0.211**| 0.214    | 0.216    | 0.393    |
>
> We will include these results in our paper.
>
> 2. We compared our results with TAPE, and the outcomes are as follows:
>
> **Table 1: The results of node-level experiments, where $ Z^{h} $ corresponds to the output variables $ \Psi_1 $,$ \Psi_2 $, and$ \Psi_3 $ of $h^{\text{node,1}}(\cdot) $, using TAPE as baseline, are presented. The results show that, due to its complexity, the TAPE method aligns with later layers of the neural network compared to those reported in the paper. We will include the relevant analysis results in our paper. Bold values indicate the minimum $ \mathcal{L}_{\text{II}} $.**
>
> |               | Layer0           | Layer1           | Layer2           | Layer3           | Layer4           |
> |---------------|------------------|------------------|------------------|------------------|------------------|
> | $\Psi_1$     | 1.620 | **1.454**  | 1.468  | 1.544  | 2.911  |
> | $\Psi_2$     | **1.276**  | 1.387 | 1.389 | 1.312  | 1.511  |
> | $\Psi_3$     | **1.380**  | 1.614  | 1.621  | 1.399 | 1.946  |
>
> **Experimental Designs Or Analyses:**
>
> We have included the experimental results for this hyperparameter and will incorporate these results into our paper:
>
> **Table 1: Accuracy corresponding to different q values, using different numbers of prompts. Llama3 was used as the enhancer. The standard deviation of results over 10 trials.**
>
> | Dataset | q=1            | q=2            | q=3            | q=4            |
> |---------|----------------|----------------|----------------|----------------|
> | Cora    | 84.94±0.81     | **86.45±0.49**     | 85.79±0.51     | 83.33±0.23     |
> | Pubmed  | 83.84±0.77     | **84.32±1.06**     | 83.98±0.39     | 82.58±0.44     |
>
> We will also provide a detailed analysis of the hyperparameters. Thank you for your suggestion.

---

> > ### Comment · Reviewer_iycs · 2025-04-07
> >
> > Thanks for your reponse, and I have no further questions.

---

### Official Review · Reviewer_2HXM · 2025-03-14

**Overall Recommendation:** 2

**Summary:**

This paper proposes a new analysis tool for LLM encoders for GNNs, based on the causal theory. The proposed method is evaluated in one synthetic dataset generated by the authors.

**Claims And Evidence:**

N.A.

**Essential References Not Discussed:**

N.A.

**Experimental Designs Or Analyses:**

Refer to "Methods And Evaluation Criteria"

**Methods And Evaluation Criteria:**

No. The proposed method is only evaluated in one synthetic dataset, which I think is not enough to show its generality and effectiveness.

**Other Comments Or Suggestions:**

N.A.

**Other Strengths And Weaknesses:**

N.A.

**Questions For Authors:**

N.A.

**Relation To Broader Scientific Literature:**

The analysis of the interpretability of LLMs encoder for GNNs is beneficial for this research area.

**Theoretical Claims:**

No. I do not have background in related theories.

---

> ### Author Rebuttal · Authors · 2025-03-31
>
> Thank you very much for your analysis and suggestions for revising our paper. We greatly appreciate your feedback. Below, we will address each of the raised concerns.
>
> **Methods And Evaluation Criteria:**
>
> Regarding the AT module, we conducted experimental analysis using several public datasets. To demonstrate the generalizability of the proposed analytical method, we first note that it can be applied to any graph dataset capable of representing higher-order causal models. Currently, validating this requires synthetic graph datasets with controllable causal relationships, where node features are rich in semantic information to effectively showcase the power of large language models. However, due to the limited availability of such datasets, we constructed our own CCSG dataset. To further establish the robustness of our approach, we expanded our evaluation by incorporating node data from the ogbn-arxiv dataset and re-executing the node-level experiments. Additionally, we conducted experiments using random data. The results are as follows:
>
> **Table 1: The results of node-level experiments, where $ Z^{h} $ corresponds to the output variables $ \Psi_1 $,$ \Psi_2 $, and$ \Psi_3 $ of $h^{\text{node,1}}(\cdot) $, using OGBN-Arxiv data and random data as node features, are presented. These results are consistent with those in the paper. Bold values indicate the minimum $ \mathcal{L}_{\text{II}} $.**
>
> |   | Layer 0             | Layer 1             | Layer 2             | Layer 3             | Layer 4             |
> |--------|---------------------|---------------------|---------------------|---------------------|---------------------|
> |$\Psi_1$     | **0.123**      | 0.223      | 0.143  | 0.134      | 0.136      |
> |$\Psi_2$    | **0.366**      | 1.337      | 0.956     | 0.998      | 1.332      |
> |$\Psi_3$     | **0.612**      | 2.747      | 2.131      | 2.324      | 2.653      |
>
> **Table 2: The results of node-level experiments, where $ Z^{h} $ corresponds to the output variables $ \Phi_1 $ and $ \Phi_2 $ of $h^{\text{node,2}}(\cdot) $, using random data as node features, are presented. These results are consistent with those in the paper. Bold values indicate the minimum $ \mathcal{L}_{\text{II}} $.**
>
> |   | Layer 0             | Layer 1             | Layer 2             | Layer 3             | Layer 4             |
> |-----------|-----------|-----------|-----------|-----------|-----------|
> |$\Psi_1$     | 1.002    | **1.202** | 1.494    | 1.625    | 2.139     |
> |$\Psi_1$     | 0.203    | **0.211**| 0.214    | 0.216    | 0.393    |
>
> We will add these results to the paper as well.

---

### Decision · Program_Chairs · 2025-05-01

**Decision:**

Accept (poster)

**Comment:**

Post-rebuttal, most reviewers (iycs, XFjz) have reached a consensus in favor of acceptance, with their main concerns (e.g., the need for broader experimental validation with additional baselines and datasets) addressed. After carefully reviewing the submission and the rebuttal, I agree that the paper presents a contribution to understanding the causality-inspired interplay between LLMs and GNNs. I recommend acceptance. Please ensure that the suggested revisions are incorporated into the final version.